# DNA Wrapping by a tetrameric bacterial histone

Yimin Hu [1], Samuel Schwab [2,6], Kaiyu Qiu [1,6], Yunsen Zhang [3], Kerstin Bär[1], Heidi Reichle[1], Aurora Panzera [4], Andrei N. Lupas [1], Marcus D. Hartmann [1,5], Remus T. Dame [2], Vikram Alva [1,7] ✉ & Birte Hernandez Alvarez [1,7] ✉

Histones are conserved DNA-packaging proteins found across all domains of life. In eukaryotes, canonical histones form octamers that wrap ~147 base pairs (bp) of DNA into nucleosomes, while in archaea they form dimers that polymerize into extended hypernucleosomes. Although bacteria were long thought to lack histones, homologs have now been identified in diverse lineages. We previously characterized the histone HBb from *Bdellovibrio bacteriovorus*, which binds and bends DNA as a dimer. Here, we describe HLp from *Leptospira perolatii* and show by crystallographic and biophysical analyses that, unlike HBb, it forms stable tetramers and binds DNA nonspecifically, wrapping ~60 bp of DNA around its core. Molecular dynamics simulations, DNA-binding assays, and heterologous expression in *Escherichia coli*, where HLp reorganizes the nucleoid, support a role in bacterial chromatin organization. These findings expand the repertoire of bacterial histone-DNA interactions and highlight the diversity of histone-based genome organization across the tree of life.

Histones are DNA-organizing proteins built around a characteristic histone fold−three α-helices (α1, α2, α3) connected by two loops−that mediates both histone-histone and histone-DNA interactions[1,2]. Histones function as oligomeric assemblies: the minimal stable unit is a dimer, which combines into tetramers, octamers, and higher-order oligomers. In eukaryotes, the canonical histone octamer forms the protein core of nucleosomes, the fundamental repeating units of chromatin. This octamer comprises a central (H3-H4)₂ tetramer (assembled from two H3-H4 dimers) flanked by two H2A-H2B dimers[1–3]. Approximately 147 base pairs (bp) of DNA wrap around this core to form individual nucleosomes arranged in a "beads-on-a-string" array, further stabilized by linker histones (H1/H5) that bind linker DNA between adjacent nucleosomes[4,5]. Specialized histone variants can deviate from canonical organization. For example, in centromeric chromatin, the histone variant CENP-A replaces H3, giving rise to CENP-A nucleosomes composed of CENP-A, H4, H2A, and H2B[6,7]. A hallmark of eukaryotic histones is their intrinsically disordered N-terminal tails, which undergo extensive post-translational modifications to regulate chromatin structure and gene expression[8,9].

Histones were long believed to be unique to eukaryotes. However, homologs have since been identified in archaea and, more recently, in bacteria, revealing that histone-based DNA organization is an ancient and widespread strategy for genome compaction[10]. Despite this shared ancestry, key differences exist across domains. While eukaryotic histones are highly conserved and structurally uniform, forming nucleosomes with defined stoichiometry, prokaryotic histones exhibit far

[1]Department of Protein Evolution, Max Planck Institute for Biology Tübingen, Tübingen, Germany. [2]Leiden Institute of Chemistry, Leiden University, Leiden, The Netherlands; Centre for Microbial Cell Biology, Leiden University, Leiden, The Netherlands; Centre for Interdisciplinary Genome Research, Leiden University, Leiden, The Netherlands. [3]Theoretical and Computational Biophysics Group, Beckman Institute for Advanced Science and Technology, Center of Biophysics and Quantitative Biology, University of Illinois Urbana–Champaign, Urbana, USA. [4]BioOptics Facility, Max Planck Institute for Biology Tübingen, Tübingen, Germany. [5]Interfaculty Institute of Biochemistry, University of Tübingen, Tübingen, Germany. [6]These authors contributed equally: Samuel Schwab, Kaiyu Qiu. [7]These authors jointly supervised this work: Vikram Alva, Birte Hernandez Alvarez. ✉e-mail: vikram.alva@tuebingen.mpg.de; birte.hernandez@tuebingen.mpg.de

greater structural and functional diversity. Notably, N-terminal tails are generally absent in prokaryotes; however, histones from Asgard archaea—the closest known prokaryotic relatives of eukaryotes—do possess such tails[11–14]. This suggests that chromatin regulation via tail modifications may have emerged prior to the evolution of the eukaryotic nucleus[13]. In both archaea and bacteria, genome organization has traditionally been attributed to nucleoid-associated proteins (NAPs), such as HU, H-NS, and IHF, which lack the histone fold but perform analogous roles in DNA compaction and transcriptional regulation[15–18]. Based on sequence analysis and structural predictions, prokaryotic histones are broadly categorized into two major families: (i) nucleosomal histones, found exclusively in archaea, and (ii) α3 histones, present in both archaea and bacteria. The α3 histones are defined by a shorter α2 helix and a truncated α3 helix, and exhibit extensive variation in their quaternary structures and domain organizations, leading to classification into multiple subfamilies[13].

Archaeal histones are more abundant and better characterized than their bacterial counterparts, particularly the nucleosomal variants. A well-established example is HMfB from *Methanothermus fervidus*, which forms homodimers that tetramerize upon DNA binding, assembling into continuous helices where each tetramer binds approximately 60 bp of DNA[19–21]. Similarly, HTkA from *Thermococcus kodakarensis* has been shown to influence chromatin compaction and significantly modulate transcription initiation and elongation[20]. Beyond the nucleosomal type, non-nucleosomal archaeal histones exhibit even greater structural diversity. Members of the face-to-face (FtF) histone subfamily—part of the α3 histone group—form homotetramers that are predicted to wrap DNA. This architecture is fundamentally distinct from the spiraling assembly seen in nucleosomal histones, as demonstrated by the crystal structure of HTkC from *T. kodakarensis*[13]. Another example, MJ1647 from *Methanocaldococcus jannaschii*, belongs to a histone subgroup found exclusively in Methanococcales. Unlike canonical histones, MJ1647 tetramerizes through its C-terminal helices rather than the histone fold and binds DNA in a bridging rather than wrapping mode[22].

Bacterial histones are rare, present in fewer than 2% of sequenced bacterial genomes, in stark contrast to the NAP HU, which is found in over 90% of genomes[23]. Their limited distribution, relatively recent discovery, and frequent occurrence in uncultured or metagenomically characterized bacteria have left most bacterial histones uncharacterized at the functional level[24]. Among those identified, two main families dominate, α3 histones and DUF1931 pseudodimeric histones, both of which also occur in archaea[13]. The DUF1931 proteins, exemplified by structures from *Thermus thermophilus* (PDB: 1WWI) and *Aquifex aeolicus* (PDB: 1R4V), form pseudodimers but lack the signature residues required for DNA binding and nucleosome-like assembly[13,25,26]. In contrast, α3 histones possess DNA-binding residues and are subdivided into five distinct subfamilies: (i) FtF histones, (ii) bacterial dimer histones, (iii) ZZ histones, (iv) phage histones, and (v) Rab GTPase histones[13]. The recent characterization of HBb from *Bdellovibrio bacteriovorus*, a member of the bacterial dimer subfamily, provided the first experimental evidence of a functional bacterial histone. HBb is essential for viability and binds genomic DNA in a sequence-independent manner as a dimer[23,27]. Although initially thought to bind without affecting DNA topology, structural and biophysical analyses later revealed that HBb induces local DNA bending, employing an interaction mode reminiscent of eukaryotic histones[27].

To broaden our understanding of bacterial histone diversity, we investigated HLp from *Leptospira perolatii*, a member of the FtF subfamily of α3 histones. Crystallographic and biophysical analyses show that HLp forms stable tetramers. Structures of HLp in both free and DNA-bound forms reveal extensive protein-DNA contacts and a tetrameric architecture resembling that of archaeal HTkC[13]. Remarkably, HLp wraps ~60 bp of DNA around its core, representing the first example of DNA wrapping by a bacterial histone. This mode of interaction, corroborated by molecular dynamics simulations and DNA-binding assays, drives DNA compaction and topological change, extending the known mechanisms of histone-DNA association in bacteria.

## Results

### Bioinformatic analysis

To explore the diversity and evolutionary relationships of bacterial FtF histones and identify candidates for experimental characterization, we performed a comparative sequence analysis based on a curated dataset comprising α3 histones from the FtF and bacterial dimer subfamilies, archaeal nucleosomal histones, and pseudodimeric DUF1931-family histones. Only proteins consisting solely of the histone fold, without additional domains, were included. All-against-all pairwise sequence similarities were analyzed using CLANS, enabling visualization of the sequence space occupied by FtF histones and assessment of their relationships to other prokaryotic histone families. The resulting cluster map reveals a clear separation among major prokaryotic histone families (Fig. 1a). DUF1931 pseudodimeric histones form a distinct, compact cluster, well separated from canonical histones. Archaeal nucleosomal histones group into a large, coherent cluster, with a small subgroup of closely related sequences extending from its core, including the *Haloferax volcanii* pseudodimer HstA (HVO_0520, UniProt D4GS56). Archaeal FtF histones, bacterial FtF histones, and bacterial dimer histones form separate, well-defined clusters. The adjacency of the bacterial FtF cluster to the archaeal FtF cluster suggests potential shared structural features, such as tetramerization.

While FtF histones are widespread across archaeal phyla, in bacteria they are predominantly found in specific lineages, including Spirochaetota, Planctomycetota, Bdellovibrionota, and Myxococcota. The majority of these bacterial representatives are derived from uncultured organisms or metagenomic assemblies, limiting opportunities for functional characterization. Among the few exceptions, we identified two species with complete genomes from culturable organisms within the bacterial FtF cluster: *L. interrogans*, a pathogenic species and the primary causative agent of leptospirosis, and *L. perolatii*, a species occasionally associated with disease in humans and animals. The FtF histone from *L. interrogans* is highly abundant and essential for viability, underscoring the functional relevance of this histone group[23]. Notably, neither *L. interrogans* nor *L. perolatii* encodes classical bacterial NAPs such as HU or Dps, and each harbors only a single histone homolog, suggesting that FtF histones may serve as the principal DNA organizers in these species. Interestingly, in many Leptospira species, the hlp gene is located adjacent to genes encoding the chromosome segregation protein SMC (WP_100713470.1), methionine aminopeptidase (MAP; WP_100713471.1), and an uncharacterized DUF350 domain-containing transmembrane protein (WP_100713472.1) (Supplementary Fig. 1). This conserved genomic neighborhood hints at potential functional links between histone-mediated DNA organization and core aspects of cellular physiology. We focused our efforts on the histone homolog HLp from *L. perolatii*, thereby enabling the characterization of a previously unexplored bacterial FtF histone and advancing our understanding of histone-based chromatin organization in Leptospira, beyond the limited insights available from *L. interrogans*[28]. Moreover, AlphaFold (AF) structure predictions indicated that HLp forms a homotetramer, resembling that of the archaeal FtF histone HTkC from *T. kodakarensis*, further supporting its relevance as a representative of this bacterial subgroup[13].

### Structure of HLp from *L. perolatii*

Based on the above bioinformatic analysis, HLp from *L. perolatii* was selected as a representative of the bacterial FtF histone group for structural and functional characterization. In sequence databases, two variants of the HLp protein are documented: a shorter form comprising 63 amino acid residues and a longer form with 77 residues,

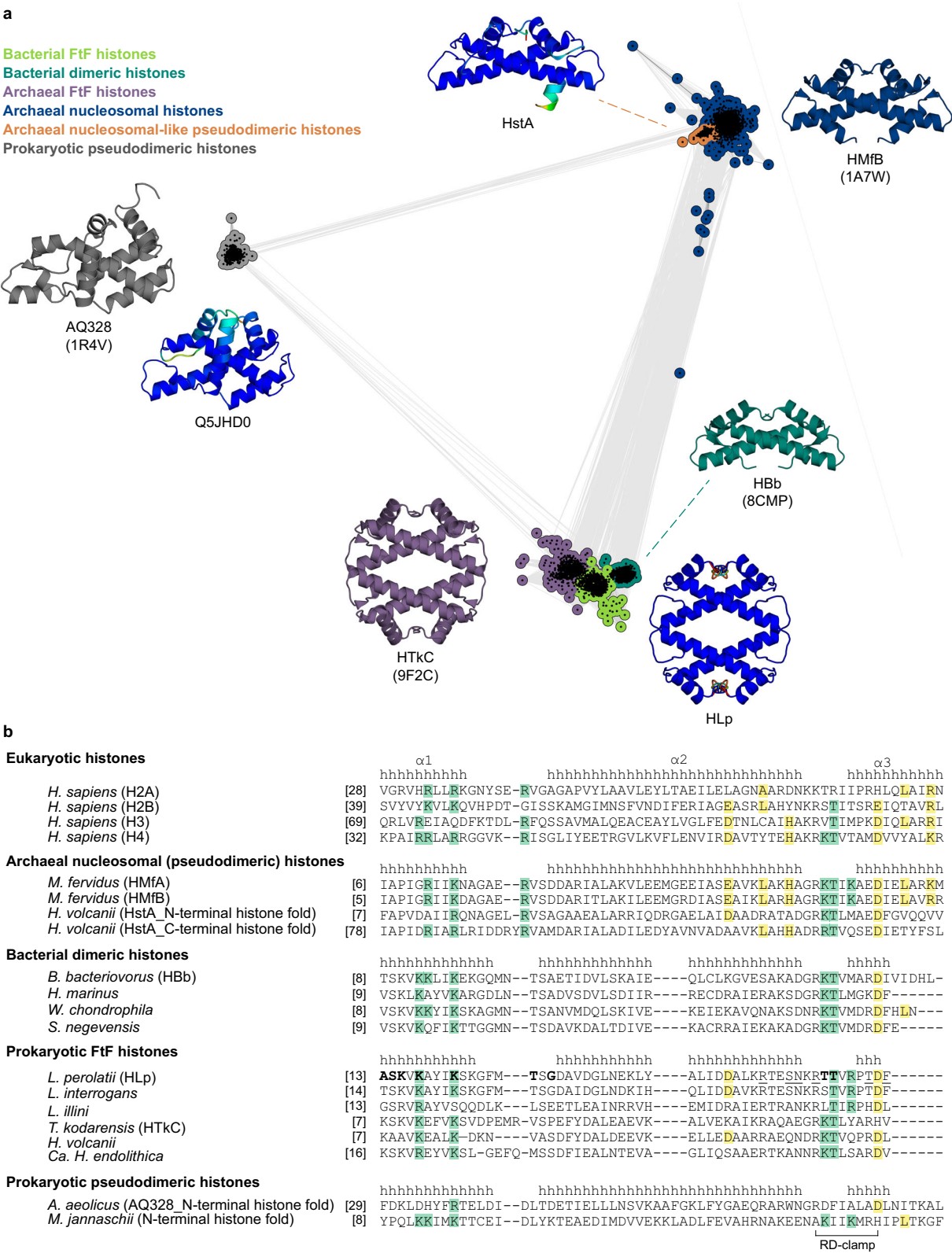

**a**

- Bacterial FtF histones
- Bacterial dimeric histones
- Archaeal FtF histones
- Archaeal nucleosomal histones
- Archaeal nucleosomal-like pseudodimeric histones
- Prokaryotic pseudodimeric histones

HstA

HMfB (1A7W)

AQ328 (1R4V)

Q5JHD0

HBb (8CMP)

HTkC (9F2C)

HLp

**b**

**Eukaryotic histones**

| | | α1 | α2 | α3 |
|---|---|---|---|---|
| | | hhhhhhhhhh | hhhhhhhhhhhhhhhhhhhhhhhhhhhhhhh | hhhhhhhhhh |
| *H. sapiens* (H2A) | [28] | VGRVHRLLRKGNYSE-RVGAGAPVYLAAVLEYLTAEILELAGNAARDNKKTRIIPRHLQLAIRN | |
| *H. sapiens* (H2B) | [39] | SVYVYKVLKQVHPDT-GISSKAMGIMNSFVNDIFERIAGEASRLAHYNKRSTITSREIQTAVRL | |
| *H. sapiens* (H3) | [69] | QRLVREIAQDFKTDL-RFQSSAVMALQEACEAYLVGLFEDTNLCAIHAKRVTIMPKDIQLARRI | |
| *H. sapiens* (H4) | [32] | KPAIRRLARRGGVK--RISGLIYEETRGVLKVFLENVIRDAVTYTEHAKRKTVTAMDVVYALK | |

**Archaeal nucleosomal (pseudodimeric) histones**

| | | hhhhhhhhhh | hhhhhhhhhhhhhhhhhhhhhhhhhhhhhhh | hhhhhhhhhh |
|---|---|---|---|---|
| *M. fervidus* (HMfA) | [6] | IAPIGRIIKNAGAE--RVSDDARIALAKVLEEMGEEIASEAVKLAKHAGRKTIKAEDIELARKM | |
| *M. fervidus* (HMfB) | [5] | IAPIGRIIKDAGAE--RVSDDARITLAKILEEMGRDIASEAIKLARHAGRKTIKAEDIELAVRR | |
| *H. volcanii* (HstA_N-terminal histone fold) | [7] | FAPVDAIIRQNAGEL-RVSAGAAEALARRIQDRGAELAIDAADRATADGRKTLMAEDFGVQQVV | |
| *H. volcanii* (HstA_C-terminal histone fold) | [78] | IAPIDRIARLRIDDRYRVAMDARIALADILEDYAVNVADAAVKLAHHADRRTVQSEDIETYFSL | |

**Bacterial dimeric histones**

| | | hhhhhhhhhhhh | hhhhhhhhhhhhh | hhhhhhhhhhhh | hhhhh |
|---|---|---|---|---|---|
| *B. bacteriovorus* (HBb) | [8] | TSKVKKLIKEKGQMN--TSAETIDVLSKAIE----QLCLKGVESAKADGRKTVMARDIVIDHL- | | | |
| *H. marinus* | [9] | VSKLKAYVKARGDLN--TSADVSDVLSDIIR----RECDRAIERAKSDGRKTLMGKDF------ | | | |
| *W. chondrophila* | [8] | VSKVKKYIKSKAGMN--TSANVMDQLSKIVE----KEIEKAVQNAKSDNRKTVMDRDFHLN--- | | | |
| *S. negevensis* | [9] | VSKVKQFIKTTGGMN--TSDAVKDALTDIVE----KACRRAIEKAKADGRKTVMDRDFE----- | | | |

**Prokaryotic FtF histones**

| | | hhhhhhhhhh | hhhhhhhhhhhh | hhhhhhhhhhhhh | hhh |
|---|---|---|---|---|---|
| *L. perolatii* (HLp) | [13] | **ASK**VKAYIKSKGFM---**T**S**G**DAVDGLNEKLY----ALIDDALKRTESNKR**TT**VRPTDF------ | | | |
| *L. interrogans* | [14] | TSKVKAYIKSKGFM---TSGDAIDGLNDKIH----QLIDDAVKRTESNKRSTVRPTDF------ | | | |
| *L. illini* | [13] | GSRVRAYVSQQDLK---LSEETLEAINRRVH----EMIDRAIERTRANKRLTIRPHDL------ | | | |
| *T. kodarensis* (HTkC) | [7] | KSKVKEFVKSVDPEMR-VSPEFYDALEAEVK----ALVEKAIKRAQAEGRKTLYARHV------ | | | |
| *H. volcanii* | [7] | KAAVKEALK-DKN----VASDFYDALDEEVK----ELLEDAARRAEQNDRKTVQPRDL------ | | | |
| *Ca. H. endolithica* | [16] | KSKVREYVKSL-GEFQ-MSSDFIEALNTEVA----GLIQSAAERTKANNRKTLSARDV------ | | | |

**Prokaryotic pseudodimeric histones**

| | | hhhhhhhhhhhh | hhhhhhhhhhhhhhhhhhhhhhhhhhhhhhhh | hhhhh |
|---|---|---|---|---|
| *A. aeolicus* (AQ328_N-terminal histone fold) | [29] | FDKLDHYFRTELDI--DLTDETIELLLNSVKAAFGKLFYGAEQRARWNGRDFIALADLNITKAL | |
| *M. jannaschii* (N-terminal histone fold) | [8] | YPQLKKIMKTTCEI--DLYKTEAEDIMDVVEKKLADLFEVAHRNAKEENAKIIKMRHIPLTKGF | |

RD-clamp

differing in their N-terminal regions (Supplementary Fig. 2). To determine the correct variant, we examined the genomic context of the hlp gene, particularly analyzing the position of the Shine-Dalgarno (SD) sequence−a ribosomal binding site crucial for initiating bacterial translation[29]. Our analysis revealed that the SD sequence aligns appropriately with the start codon of the shorter, 63-residue variant, suggesting that this form is the authentic translation product. HLp was recombinantly overexpressed in *E. coli* and purified to homogeneity, as confirmed by SDS-PAGE, which showed a single band at ~10 kDa, consistent with its theoretical molecular weight of 7.1 kDa. SEC-MALS revealed a predominant species of 26 ± 0.3 kDa, indicating that HLp forms tetramers in solution. CD spectroscopy showed that HLp is a predominantly α-helical protein and unfolds upon heating, with a melting temperature of 57.1 °C (Supplementary Fig. 3).

**Fig. 1 | Comparative analysis of prokaryotic histones composed exclusively of histone fold domains. a** Cluster map of ~3300 prokaryotic histones containing only the histone fold domain. Each dot represents a single protein sequence, edges reflect pairwise sequence similarity. Structures of representative proteins—either crystal structures (PDB IDs in parentheses) or AlphaFold models—are shown in cartoon representation. AlphaFold models are colored by pLDDT confidence scores. Representative proteins include: HstA from *Haloferax volcanii* (D4GS56), an archaeal nucleosomal-like pseudodimeric histone; HMfB from *Methanothermus fervidus* (P19267), an archaeal nucleosomal histone; HBb from *Bdellovibrio bacteriovorus* (Q6MRM1), a bacterial dimeric histone; HLp from *Leptospira perolatii* (A0A2M9ZN55), a bacterial FtF histone; HTkC from *Thermococcus kodakarensis* (Q5JDW7), an archaeal FtF histone; and Q5JHD0 from *T. kodakarensis* (Q5JHD0) and AQ328 from *Aquifex aeolicus* (O66665), both prokaryotic pseudodimeric histones. **b** Multiple sequence alignment of representative histones from both eukaryotes

and prokaryotes, including *Homo sapiens* (H2A: P04908; H2B: P62807; H3: P68431; H4: P62805), *M. fervidus* (HmfA: P48781; HMfB: P19267), *H. volcanii* (HstA: D4GS56), *B. bacteriovorus* (HBb: Q6MRM1), *Hymenobacter marinus* (NCBI accession No.: ATH09486), *Waddlia chondrophila* (D6YWW1), *Simkania negevensis* (F8L7X8), *L. perolatii* (HLp: A0A2M9ZN55), *L. interrogans* (Q8F3E8), *Leptonema illini* (H2CFS2), *T. kodakarensis* (HTkC: Q5JDW7), *H. volcanii* (D4GZE0), Candidatus *Heimdallarchaeum endolithica* (A0A9Y1FPJ9), *A. aeolicus* (O66665), and *Methanocaldococcus jannaschii* (A0A832T4V6). α-helices are annotated with "h" based on crystal structures of H2A, HMfA, HBb, HLp, and AQ328. Conserved residues associated with DNA binding and oligomerization are highlighted in green and yellow, respectively. HLp residues contributing to tetramerization in the HLp crystal structure (PDB: 9QT0) are underlined, while those interacting with the DNA backbone in HLp-DNA complexes (PDB: 9QT1 and 9QT2) are shown in bold. Unless otherwise stated, accession numbers in parentheses refer to UniProtKB.

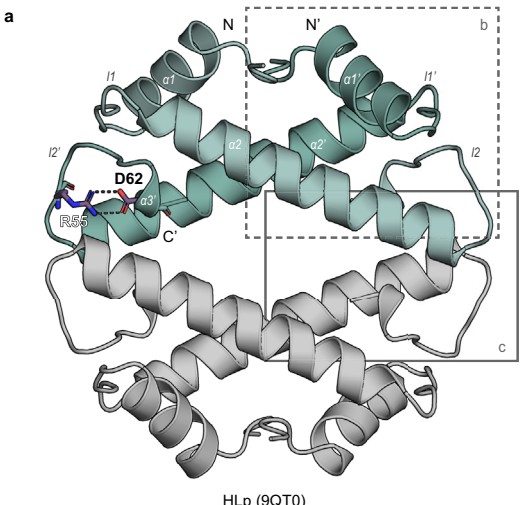
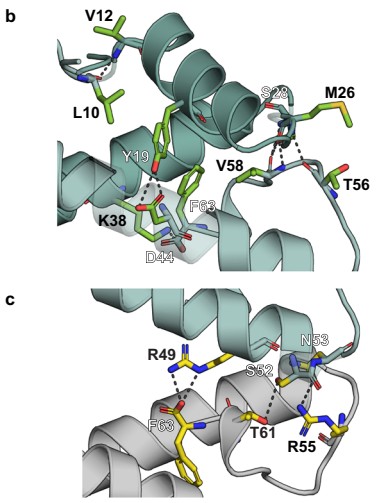

**Fig. 2 | Crystal structure of the HLp tetramer. a** Crystal structure of the HLp tetramer (PDB: 9QT0) shown in cartoon representation. Residues R55 and D62, which form the RD clamp, are shown as sticks. Salt bridges are indicated as dashed lines. **b** Close-up view of the HLp dimerization interface. Residues involved in dimerization are shown as green sticks, with hydrogen bonds depicted as dashed lines. **c** Close-up view of the HLp tetramerization interface. Tetramerization-associated residues are shown as yellow sticks. Salt bridges and hydrogen bonds are indicated by dashed lines. In all panels, the contents of the asymmetric unit are shown in color, with selected symmetry mates in gray.

HLp crystallized under multiple conditions within a few days. The best data set was processed to a resolution of 1.30 Å, and the structure was solved by molecular replacement using a dimeric AF model as the search template (Fig. 2)[30,31]. The entire chain, except for the first seven residues, was well resolved in the electron density map (Supplementary Fig. 4a). The asymmetric unit contained an HLp dimer, which assembles into a homotetramer by crystallographic symmetry. This tetramer closely matches the AF model, with a Root Mean Square Deviation (RMSD) value of 0.907 Å (Supplementary Fig. 5a).

The HLp monomer exhibits the characteristic histone fold, comprising three α-helices (α1, α2, and α3) connected by two loops (l1 and l2) (Fig. 2a). The central α2 helix is one turn shorter than in the archaeal histone HMfB (Supplementary Fig. 6), while the C-terminal α3 helix is truncated and forms a single helical turn composed of the final four residues—similar to what is observed in the bacterial dimeric histone HBb from *B. bacteriovorus*[27]. Within the dimer, the monomers are arranged in a head-to-tail orientation, with the interface formed primarily by the antiparallel crossing of the α2 helices at an angle of ~40°. It is stabilized by hydrophobic interactions involving residues V16 and I20 (α1 helix), a leucine-rich stretch in the α2 helix (L35, L39, L42, and L47), and F63 (α3 helix), along with polar contacts between loop regions. Notably, M26 and S28 in l1 of one monomer interact with T56 and V58 in l2 of the opposing monomer (Fig. 2b).

HLp tetramers assemble through two equivalent, spatially separated interfaces formed by the C-terminal α2 helices, l2 loops, and the α3 helices of opposing dimers. Dimer-dimer contacts are stabilized by a branched hydrogen-bonding network involving R49, S52, and N53 of one monomer and F63, T61, and R55 of the opposing monomer (Fig. 2c). R49, N53, and R55 are conserved among predicted tetrameric bacterial FtF histones, indicating a shared tetramer-assembly mechanism (Fig. 1b and Supplementary Fig. 7a, b). By contrast, eukaryotic and archaeal nucleosomal tetramers interact through a single interface stabilized by a four-helix bundle and display a different relative orientation of the two dimers (Supplementary Fig. 7c, d)[2,19].

HLp also harbors a conserved RD-clamp—an intramolecular salt bridge between R55 and D62—that stabilizes loop l2, a feature shared with both HMfB and HBb (Figs. 1b, 2a and Supplementary Fig. 6)[27,32]. Compared to canonical histones, HLp shows partial conservation of residues implicated in DNA binding (Fig. 1b). Nevertheless, electrostatic surface potential analysis of the HLp tetramer reveals a continuous band of positive charge encircling the structure, consistent with a DNA-wrapping mode of interaction (Supplementary Fig. 5b)[33].

## HLp binds to DNA in vitro

We first assessed the DNA-binding ability of HLp using EMSAs. An 80-bp (~50% GC) DNA fragment—previously shown to bind HMfB with high affinity—exhibited reduced mobility in polyacrylamide gels upon

incubation with HLp (Fig. 3a), indicating the formation of HLp-DNA complexes[19,34]. Complex formation was concentration dependent, and the appearance of smeared bands suggested that the HLp-DNA complexes are less stable or more heterogeneous than those formed with HMfB.

To probe sequence preferences, we performed EMSAs with shorter 30-bp DNA fragments varying in GC content (30, 40, 50, and 60%). These assays produced sharper band patterns and revealed that HLp binds most strongly to the 40% GC fragment (Fig. 3b and Supplementary Fig. 8). We then used microscale thermophoresis (MST) to compare the binding affinities of HLp and HMfB with both the 80-bp fragment and the 80-bp-GC40 fragment. Both histones displayed dissociation constants in the low micromolar range. Whereas HMfB bound the two fragments with similar affinity, HLp showed a modest (~ two-fold) preference for the 80-bp-GC40 fragment in MST measurements (Fig. 3c and Supplementary Table 2). The observed differences in affinity most likely reflect sequence-dependent structural features of the DNA substrates. Finally, we used SEC-MALS to determine the stoichiometry of the HLp-DNA complex. HLp alone and the 30-bp-GC40 DNA fragment each eluted as single peaks with molecular weights matching their theoretical values (Fig. 3d). Upon incubation of HLp with 30-bp-GC40 DNA, a new peak appeared with an apparent molecular weight of 69.2 ± 5.7 kDa, consistent with a complex consisting of an HLp tetramer bound to two 30-bp DNA duplexes.

## Crystal structures of HLp bound to DNA

To gain structural insight into the interaction between HLp and DNA, we co-crystallized HLp and the 30-bp-GC40 fragment. Diffraction data were collected for two different crystal forms, HLp-DNA_1 and HLp-DNA_2, processed to resolutions of 2.10 Å and 1.90 Å, respectively (Supplementary Table 3). Both structures were solved by molecular replacement using the DNA-free HLp structure as the search model. In both HLp-DNA crystal forms, the DNA fragments appear to form continuous double helices winding throughout the crystal lattice, each revealing a distinct interface between HLp and the DNA (Fig. 4a, b). In both structures, the seemingly continuous DNA bases within central segments (16-bp dsDNA for HLp-DNA_1 and 15-nt ssDNA for HLp-DNA_2) were well resolved in the electron density maps, whereas the first six N-terminal residues of HLp were not visible (Supplementary Fig. 4b, c).

In HLp-DNA_1, the asymmetric unit contains an HLp dimer, which assembles into tetramers by crystallographic symmetry, and a 16-bp dsDNA segment of the seemingly infinite DNA helices winding throughout the crystal (Fig. 4a). As observed in other histone-DNA complexes, binding is mediated primarily through interactions between basic or polar side chains of HLp and the phosphate backbone of the DNA. On HLp, the interface spans the two monomers in the ASU, across the dimer. In each monomer, this interface involves residues A13, S14, and K15, which correspond to the "paired end of helices" motif in eukaryotic histones, as well as K21 in helix α1, and T27 and G29 in loop l1 (Fig. 4c, e)[35].

In HLp-DNA_2, the ASU comprises a single HLp monomer, which forms tetramers via crystallographic symmetry, and a 15-nt ssDNA segment of the apparently endless DNA (Fig. 4b). Although the overall crystal packing is similar to HLp-DNA_1, it exhibits a complementary DNA binding mode, with the dsDNA engaged along the HLp tetramerization interface. Therein, in both HLp dimers, the monomers bind to DNA via two sets of interactions. The first corresponds to the "β-bridge" motif in eukaryotic histones and involves T27 and G29 in loop l1, as well as T56 and T57 in loop l2, while the second set involves K17 and K21 from helix α1 (Fig. 4d, f)[35].

The two DNA-bound crystal structures thereby reveal complementary, but overlapping DNA-binding interfaces distributed along the orbicular surface of the HLp tetramer, corresponding to the canonical interaction motifs of archaeal and eukaryotic histones[19,33,35].

Taken together, these findings suggest that HLp is able to bind DNA through a wrapping mechanism.

## Molecular dynamics (MD) simulation

The two crystal structures suggested a bridging mode of DNA binding by HLp. However, given the distinct yet overlapping interfaces observed, we hypothesized that HLp might also support a wrapping mode, in which the tetramer engages a DNA segment across its entire DNA-binding interface. To investigate whether HLp indeed exhibits both bridging and wrapping modes, and to explore their relative stability and potential interconversion, we performed all-atom MD simulations. Using the HLp-DNA_1 and HLp-DNA_2 structures as templates, we constructed two DNA-binding models: (i) a wrapping model, consisting of an HLp tetramer encircled by a 67-bp dsDNA fragment, and (ii) a bridging model, in which the tetramer engages two separate 32-bp dsDNA fragments (Fig. 5a). For each mode, we generated two independent starting conformations, derived from HLp-DNA_1 and HLp-DNA_2, respectively, and ran 1 μs all-atom simulations for each (Supplementary Figs. 9, 10 and Supplementary Movies 1, 2). As the paired trajectories for each mode displayed highly consistent behavior, one representative trajectory was selected for detailed analysis (Fig. 5b), while the other is provided for comparison in the Supplementary Information (Supplementary Fig. 11).

The wrapping model remained stable throughout the simulation, maintaining a DNA backbone RMSD of approximately 6 Å (Fig. 5b). In the final frame, the HLp tetramer was fully wrapped by ~60 bp of dsDNA (Fig. 5a). By contrast, the bridging model underwent substantial conformational rearrangements, including a progressive increase in protein-DNA contacts and a spontaneous transition toward a wrapping configuration (Fig. 5a, b). Free-energy landscape analysis corroborated this trend: the wrapping model exhibited a deeper, more localized minimum basin, indicating a more convergent conformational ensemble (Supplementary Fig. 12). Importantly, in both models the HLp tetramer remained structurally stable throughout the simulations, as evidenced by the consistent planarity of the tetramer measured via the dihedral angle between its two dimers (Supplementary Fig. 13a).

To quantify HLp-DNA contacts, we calculated atom-atom interactions between protein and DNA heavy atoms for both the bridging and wrapping models (Supplementary Fig. 13b, c). Most contacts involved the DNA phosphate backbone and sugar moieties, consistent with the non-sequence-specific binding observed in crystal structures. HLp side chains contributed slightly more to DNA interactions than backbone atoms. Across both models, residues located in α1 and loops l1 and l2 exhibited an average of more than three DNA contacts (Supplementary Figs. 14, 15). In particular, K54 and R59 in loop l2 formed persistent hydrogen bonds with the DNA throughout the simulations (Supplementary Fig. 16).

To dissect the spatial dynamics of binding, we defined four major DNA-binding regions on the HLp tetramer: A-sites 1 and 2, each comprising two "β-bridge" motifs with their flanking DNA-binding residues; and B-sites 1 and 2, each centered on a "paired end of helices" motif along with adjacent DNA-interacting residues (Fig. 5c and Supplementary Table 4). In the bridging model, DNA remained stably associated with B-sites 1 and 2, but progressively engaged A-site 2, and to a lesser extent A-site 1, as it transitioned into the wrapping mode (Fig. 5d and Supplementary Fig. 17). In the wrapping model, local interactions at all four sites were stably maintained or even enhanced over time (Fig. 5d and Supplementary Fig. 18). Notably, fluctuations in contact number at A-site 1 in both models appear to result from steric interference at the DNA termini.

To probe the unwrapping dynamics of the HLp-DNA complex, we performed steered molecular dynamics (SMD) simulations (Supplementary Fig. 19). Unwrapping occurred in two distinct phases: initial disruption at B-site 2, followed by disengagement at A-site 2, yielding a

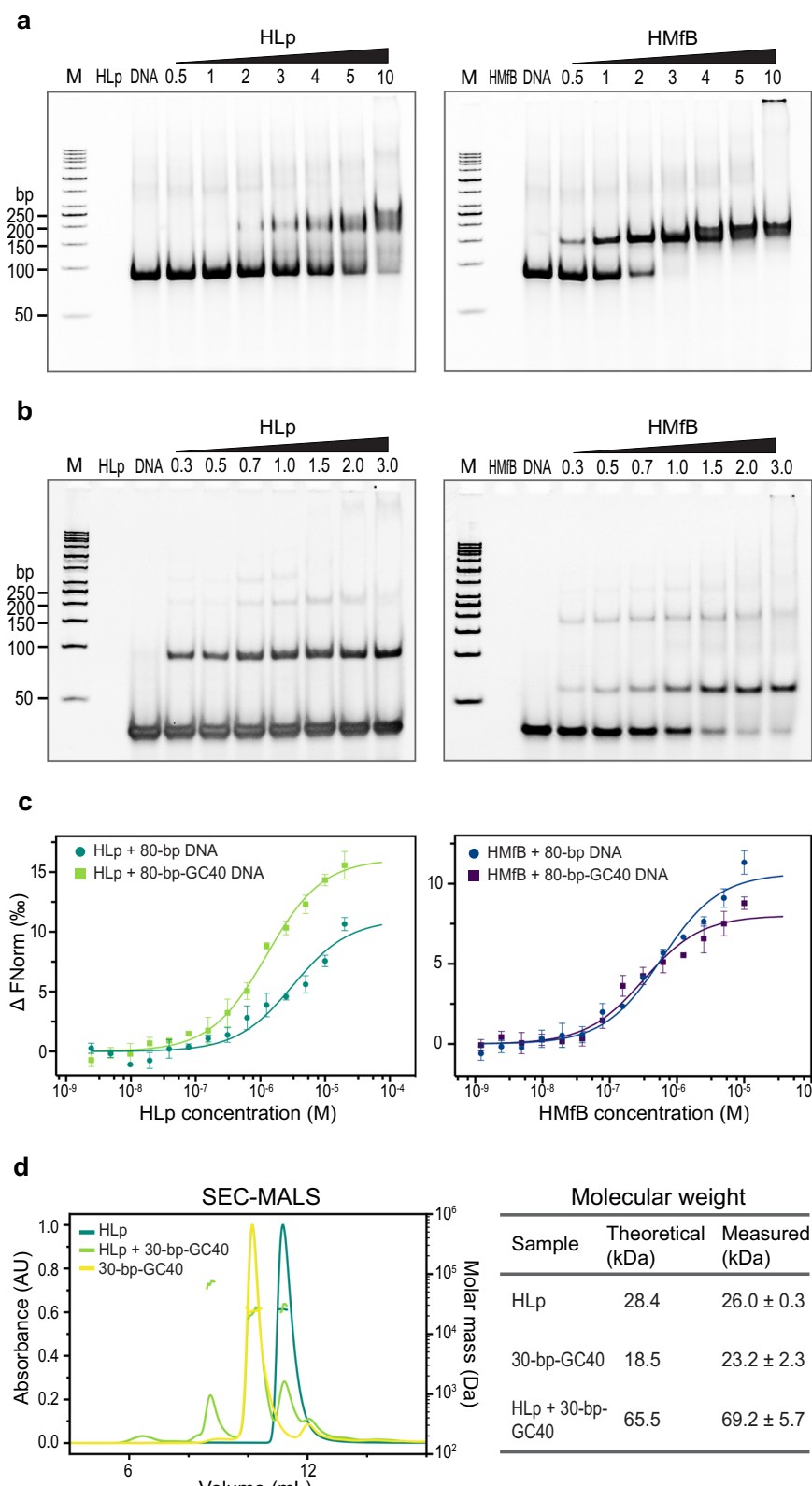

**Fig. 3 | HLp binds DNA in vitro. a** EMSAs showing the binding of HLp and the control HMfB (control) to the 80-bp DNA fragment. **b** EMSAs showing the binding of HLp and HMfB (control), the 30-bp-GC40 DNA fragment. Increasing protein concentrations (lanes 4–10), indicated as molar protein-to-DNA ratios, were incubated with the corresponding DNA and analyzed on a 6% polyacrylamide gel. Experiments in (**a**) and (**b**) were performed independently three times with similar results. **c** MST measurements comparing the binding of HLp to the 80-bp and the 80-bp-GC40 DNA fragments, alongside HMfB. Each data point represents the mean of three independent measurements from different protein batches, with error bars indicating the standard deviation. **d** SEC-MALS profiles from runs of HLp, 30-bp-GC40 DNA, and their mixture are shown. The table lists both theoretical and experimentally determined molecular weights; the latter correspond to the mean of three replicates of the same protein preparation. Source data are provided as a Source Data file.

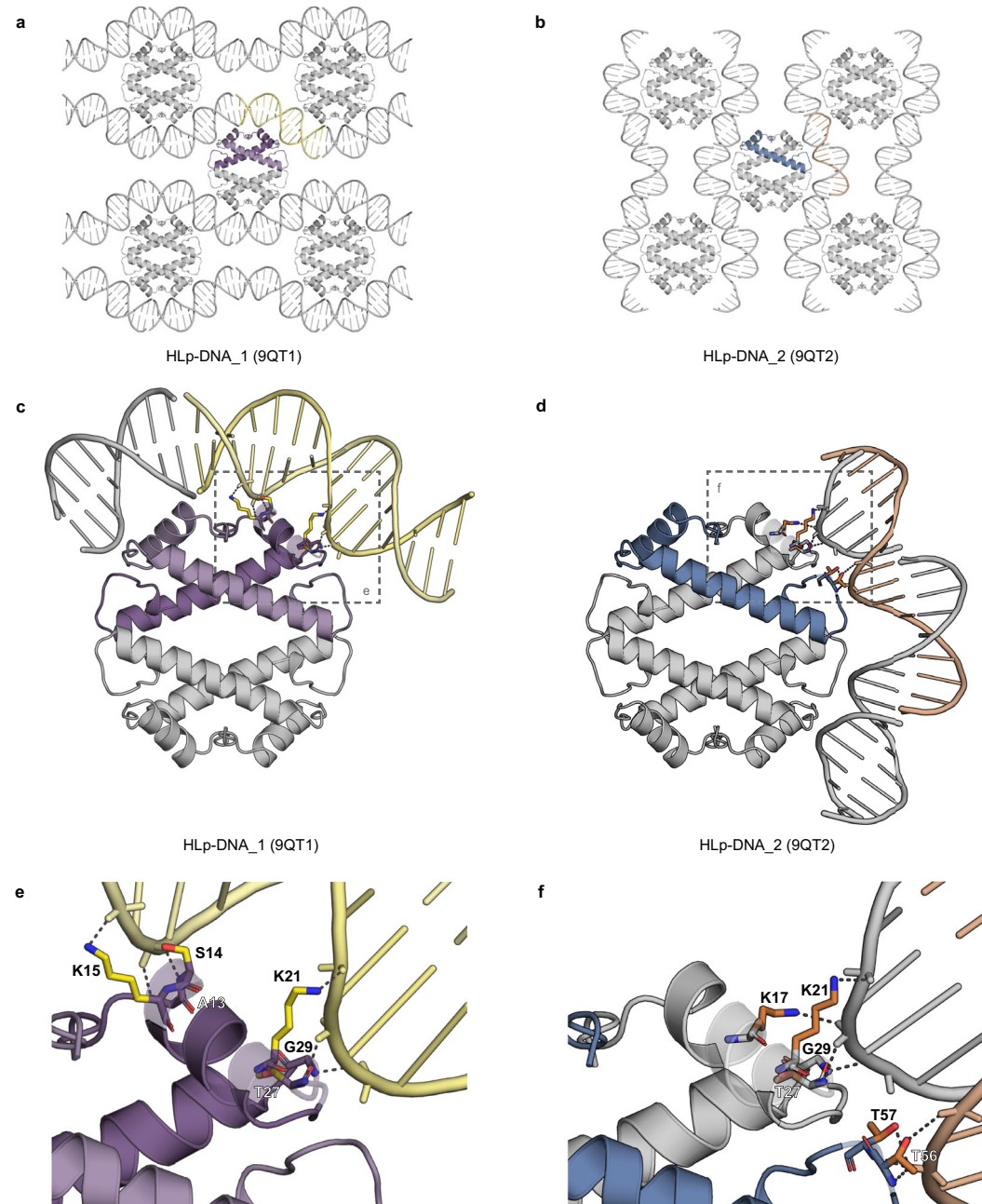

**Fig. 4 | Crystal structures of DNA-bound HLp. a** Crystal packing of HLp-DNA_1 showing selected symmetry mates within 20 Å. **b** Crystal packing of HLp-DNA_2 with selected symmetry mates within 20 Å. **c** Crystal structure of HLp-DNA_1 (PDB: 9QT1) shown in cartoon representation. **d** Crystal structure of HLp-DNA_2 (PDB: 9QT2) in cartoon representation. **e**, **f** Magnified views of the framed regions in panels (**c**, **d**) respectively. Residues involved in DNA binding are shown as sticks, and protein-DNA interactions are depicted as dashed lines. In all panels, the contents of the asymmetric unit are shown in color, and selected symmetry mates in gray.

partially unwrapped intermediate ensemble. The unwrapping energy landscape revealed two main barriers: ~500 kJ/mol at B-site 2 and ~400 kJ/mol at A-site 2. Energy decomposition analysis identified five residues—R59, K21, K15, S14, and K17—as major contributors to DNA binding, collectively accounting for 52% of the total binding energy. These findings are consistent with the hydrogen bond and contact analyses from the unbiased simulations and highlight the cooperative, multivalent nature of HLp-DNA interactions. The high energetic cost of unwrapping suggests that dissociation is a regulated, stepwise process, reinforcing the stability of the wrapped state.

Taken together, our simulations support both wrapping and bridging as viable DNA-binding modes for HLp. Nevertheless, the wrapping mode sampled a more convergent conformational ensemble and maintained more persistent protein-DNA contacts, suggesting that it is likely to predominate under physiological conditions.

## HLp is wrapped by DNA in vitro

To investigate the mechanism of DNA binding by HLp in vitro, we employed a series of well-established biochemical and biophysical assays. We first used an MNase digestion assay, which measures the extent to which protein-bound DNA is protected from nucleolytic cleavage. The 600-bp dsDNA fragment was incubated with HLp and subsequently digested with increasing concentrations of MNase. The resulting digestion pattern revealed a characteristic ladder of

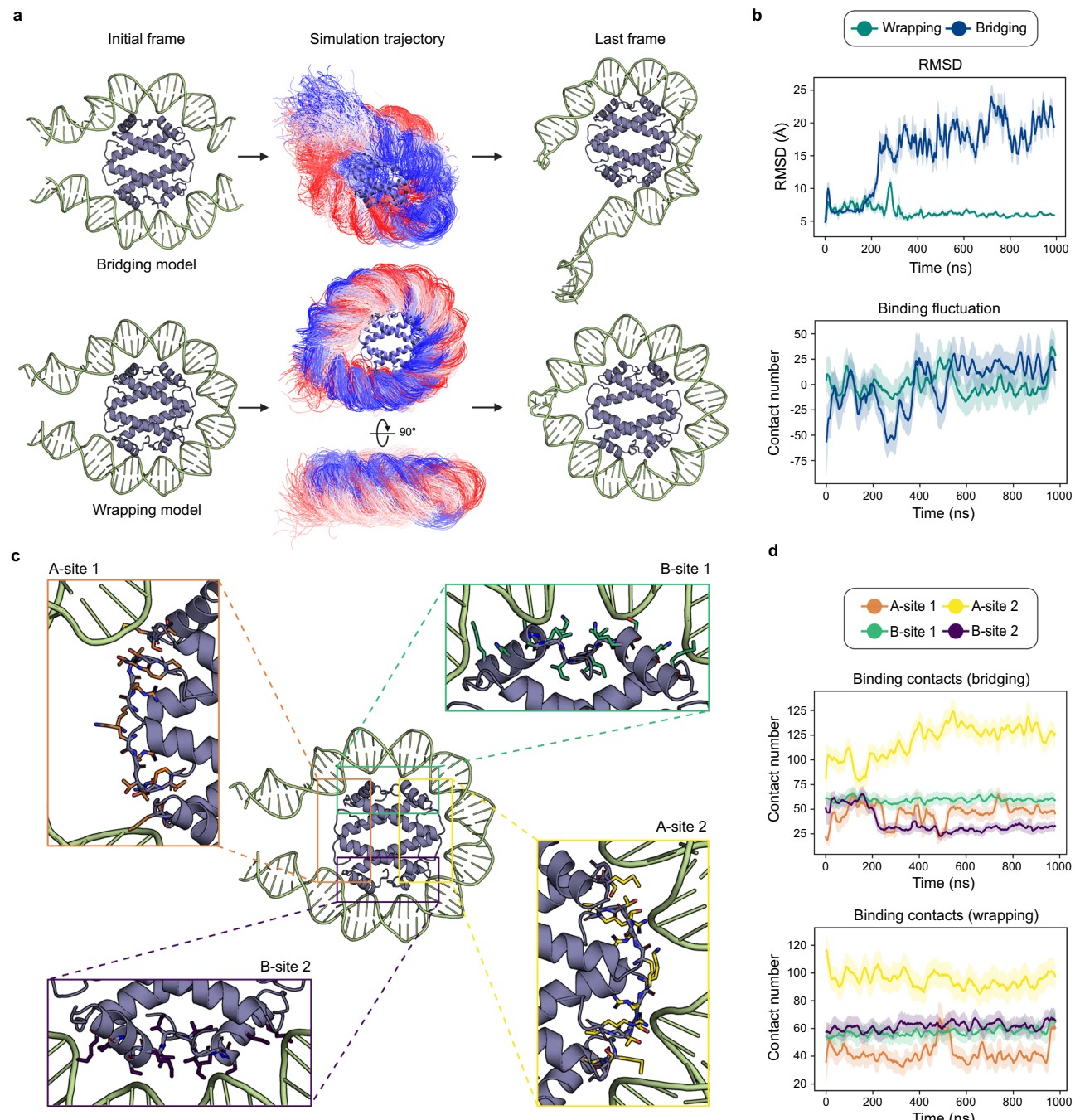

**Fig. 5 | Molecular dynamics simulations of the wrapping and bridging modes.**
**a** Visualization of the starting model (left), the conformational ensemble derived from the simulation (middle), and the structure of the last frame (right). **b** Overall characterization of the simulations of the two proposed binding modes, showing the RMSD (Root Mean Square Deviation) of the DNA backbones (top) and binding fluctuations (bottom). Binding fluctuation is defined as the deviation of the contact number in each frame from the average contact number across all frames. Each plot is smoothed with a window size of 100, with green and blue lines representing the wrapping and bridging models, respectively. **c** Structures of the four defined binding sites: A-site 1 (orange), A-site 2 (yellow), B-site 1 (green), and B-site 2 (purple). Residues involved in each site are shown as sticks and listed in Supplementary Table 4. **d** Binding contacts for the four defined binding sites throughout the simulation for the bridging (top) and wrapping (bottom) models. The graph displays the mean number of binding contacts (line) with the standard deviation (error bands) across all frames. Line colors correspond to those in panel (**c**). Binding contacts are defined as any pair of heavy atoms within 4 Å.

fragments ranging from 35 to 72 bp (Fig. 6a), indicating that HLp protects bound DNA from enzymatic degradation. This protective effect is reminiscent of that observed for the nucleosomal archaeal histone HMfB, which served as a positive control. However, the HLp-protected fragments differ in both size and regularity: while HMfB consistently generates ~30-bp increments, the fragment sizes protected by HLp are more heterogeneous. These observations suggest

that although both proteins compact DNA, they do so through distinct binding modes.

To directly monitor HLp-induced DNA compaction, we performed TPM experiments using the 685-bp linear DNA fragment. TPM enables real-time monitoring of protein-DNA interactions by measuring the Brownian motion of a DNA-tethered polystyrene bead; changes in DNA flexibility or contour length, such as those caused by protein-induced

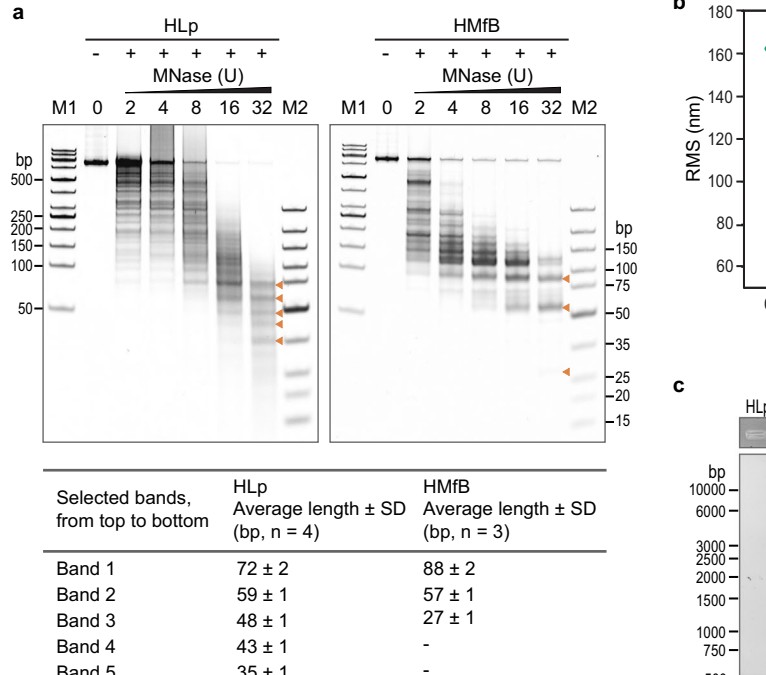

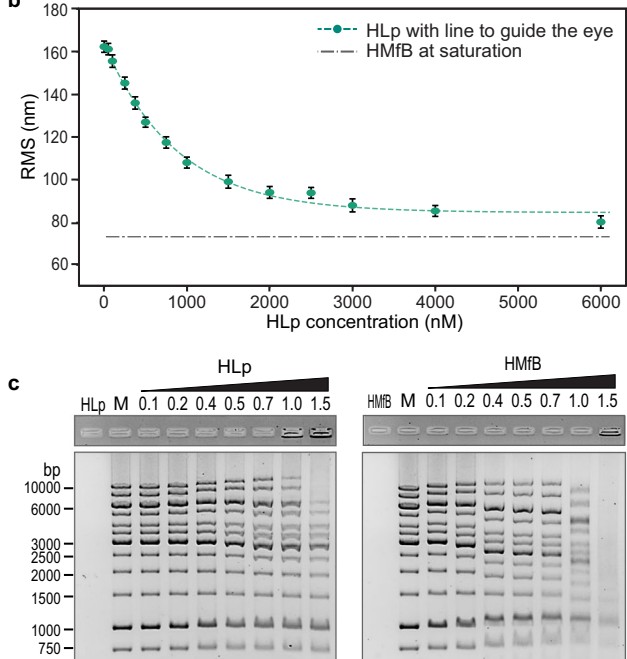

| Selected bands, from top to bottom | HLp Average length ± SD (bp, n = 4) | HMfB Average length ± SD (bp, n = 3) |
|---|---|---|
| Band 1 | 72 ± 2 | 88 ± 2 |
| Band 2 | 59 ± 1 | 57 ± 1 |
| Band 3 | 48 ± 1 | 27 ± 1 |
| Band 4 | 43 ± 1 | - |
| Band 5 | 35 ± 1 | - |

**Fig. 6 | DNA wraps around HLp. a** MNase digestion assay analyzing the protection of the 600-bp-GC40 DNA fragment by HLp in comparison to HMfB. Increasing amounts of MNase used in the assay are indicated. The table summarizes DNA fragment lengths obtained at the highest amount of MNase, determined by densitometric analysis. The densitometric analysis was performed for the indicated number (n) of independent experiments, and the average DNA fragment lengths with standard deviations (SD) are listed in the accompanying table. The molecular weight marker M2 (GeneRuler Ultra Low Range DNA Ladder, ThermoFisher Scientific) was used as the reference. **b** TPM experiment with HLp and the 685-bp DNA.

Each measurement point represents the average of three technical replicates, with error bars indicating standard deviations. The connecting line was obtained by fitting the means to a logistic function. RMS refers to the root mean square displacement. **c** EMSA analyzing the binding of HLp and HMfB (control) to the DNA fragments of the GeneRuler 1 kb Ladder (ThermoFisher Scientific). Increasing protein concentrations (lanes 3–9), indicated as protein-to-DNA mass ratios, were incubated with the corresponding DNA and analyzed on a 1% agarose gel. The experiment in (**c**) was performed independently three times for HLp and twice for HMfB, yielding similar results. Source data are provided as a Source Data file.

compaction, result in measurable shifts in root mean square (RMS) displacement of the bead. Upon addition of HLp, we observed a progressive, concentration-dependent decrease in RMS values, indicating compaction of the DNA (Fig. 6b). Saturation was reached at ~6000 nM HLp with a final RMS value of ~80 nm, closely matching the values observed for the archaeal histones HMfA and HMfB, suggesting that HLp wraps DNA rather than bridging it ref. [36].

To investigate whether HLp induces DNA condensation across a range of DNA sizes, we performed ladder EMSA assays with linear dsDNA fragments of increasing length. HLp altered the electrophoretic mobility of DNA on agarose gels in a size-dependent manner, with fragments longer than 2000 bp migrating faster, while those shorter than 1000 bp showed reduced mobility (Fig. 6c). These results are consistent with protein-induced changes in DNA conformation and are qualitatively similar to those observed with HMfB, although HMfB required lower protein concentrations and produced more pronounced mobility shifts[21].

We next examined whether HLp binding affects DNA topology using a topoisomerase I relaxation assay. In the presence of HLp, the relaxed plasmid DNA became progressively supercoiled in a protein concentration-dependent manner, indicating the introduction of topological strain upon HLp binding (Supplementary Fig. 20a). This behavior mirrors that of HMfB, albeit with a less pronounced effect.

Finally, we tested whether HLp promotes DNA end-joining using a ligase-mediated circularization assay. In this assay, short DNA fragments are circularized by T4 DNA ligase and non-circularized linear species are digested by T5 exonuclease. While HMfB efficiently promotes the circularization of DNA monomers due to its pronounced DNA-bending activity, HLp had only modest effects, even at high concentrations (Supplementary Fig. 20b). In contrast, HLp favored the formation of linear DNA multimers as protein concentration increased, indicative of open-ended protein-DNA complexes formed upon binding.

Taken together with the crystal structures and molecular dynamics simulations, our in vitro data support a model in which HLp wraps and compacts DNA through a mechanism that is distinct from the bending mode employed by the bacterial histone HBb and from the nucleosome assembly of eukaryotic histones, differing in both its dynamics and topological effects.

## HLp binds to genomic DNA in vivo
Due to the lack of facilities for handling Leptospira strains, we used *E. coli* as a heterologous model to investigate the effects of HLp on genomic DNA in vivo using light microscopy. HLp was expressed in *E. coli* BL21(DE3), while the archaeal histone HMfB was expressed in *E. coli* Mutant56(DE3) as a positive control. Ubiquitin from *Caldiarchaeum subterraneum* (*Cs*Ub), which lacks DNA-binding activity, served as a negative control[37]. Because expression levels varied significantly between constructs, the strain showing the highest expression level for each protein was selected for further analysis.

Growth curves were recorded to determine the onset of the stationary phase and to assess potential effects of protein expression on cell proliferation (Supplementary Fig. 21). HLp and *Cs*Ub expression in *E. coli* BL21(DE3) led to a modest increase in doubling time and a reduction in final cell density compared to uninduced controls. However, similar effects were observed in cells carrying empty vectors, suggesting that these changes were not specific to HLp or *Cs*Ub expression. In contrast, HMfB expression had no measurable impact

on cell growth. To minimize variability due to nucleoid dynamics during the cell cycle, microscopy samples were collected in early stationary phase, when cells had exited active division.

Microscopic analysis revealed that HLp expression markedly alters nucleoid organization. DAPI staining of HLp-expressing cells demonstrated clear elongation and a substantial increase in nucleoid volume relative to controls (Fig. 7), consistent with impaired chromosome condensation and/or interference with cell division. These effects were already evident in the absence of IPTG, likely due to basal expression from the leaky T7 promoter. Upon expression of HLp, the cytoplasm of HLp-expressing cells was predominantly occupied by decondensed genomic DNA. This phenotype was also observed upon HMfB expression, suggesting that HLp disrupts the chromatin organization of the *E. coli* cells through its DNA-binding ability. Segmentation of the DAPI fluorescence signals in HMfB-expressing cells exhibited single, well-defined regions with clear contours. In contrast, segmentation of HLp-expression cells resulted in fluorescence signals dispersed across multiple regions with diffuse contours, indicating a more heterogeneous distribution of these nucleoid regions.

To validate the HLp-DNA interaction in vivo, we performed an MNase digestion assay of genomic DNA from *E. coli* expressing either HLp or HMfB (Fig. 8). In both cases, MNase-resistant ladders were observed, indicating protection of chromosomal DNA. The ladder produced by HLp was distinct from that generated by HMfB, yet it closely resembled the fragment pattern obtained in our in vitro MNase digestion using the 600-bp-GC40 DNA fragment, supporting the conclusion that HLp wraps DNA in vivo and aligning with our structural and biochemical data.

Combined with the TPM assay results, which showed HLp-induced DNA compaction only at high protein concentrations, these findings suggest that HLp induces localized DNA condensation and reveal subtle differences in chromatin organization between the two histones.

## Discussion

In this study, we present a comprehensive analysis of the bacterial histone HLp from *L. perolatii*, integrating structural, biophysical, and functional data to elucidate its role in DNA binding and compaction. Our findings reveal that HLp assembles into a unique, DNA-wrapped tetramer, expanding the known repertoire of bacterial histone architectures and providing insights into prokaryotic chromatin organization. This represents the first experimental characterization of a bacterial histone belonging to the FtF subgroup of α3 histones, previously identified bioinformatically as one of the largest subfamilies of prokaryotic histones[13].

Structurally, HLp closely resembles the archaeal FtF histone HTkC from *T. kodakarensis*, consistent with bioinformatic predictions based on CLANS clustering, where the two histones are positioned adjacent to each other. Both proteins adopt a characteristic histone fold comprising three α-helices connected by two loops, yet differ notably from nucleosomal histones in their truncated α2 and α3 helices. This truncation is also found in HBb, the only other experimentally studied bacterial histone, highlighting a common structural adaptation within α3 histones[23,27]. Unlike HBb, which strictly forms dimers, HLp and HTkC assemble into tetramers even in the absence of DNA, mediated by highly conserved residues situated at the C-terminal region of α2, loop l2, and the helix α3. This unique oligomerization interface appears to be a defining feature of FtF histones and sets them apart from other prokaryotic histone families, including archaeal nucleosomal histones, which typically assemble into higher-order oligomers only upon DNA binding. Given their structural similarity and conserved tetramerization interface, we anticipate that HTkC may also wrap DNA similarly to HLp.

Tetramer formation occurs in both prokaryotic and eukaryotic histones, but these tetramers have evolved distinct properties tailored to their respective cellular functions. In eukaryotes, at least two types of histone tetramers exist: (H3-H4)$_2$ heterotetramers, which form the stable structural core of canonical nucleosomes, and CENP-A/H4 tetramers, which assemble into centromere-specific nucleosomes in which CENP-A replaces H3[2,6,7,35]. Both tetramer types are equally stabilized by a four-helix bundle at their dimer–dimer interface, yet they differ markedly in mechanical behavior: unlike (H3-H4)$_2$ tetramers, CENP-A tetramers exhibit pronounced structural plasticity, with MD simulations showing that shear along the interface can even reverse the handedness of bound DNA[38,39]. This distinctive property likely underlies cell cycle-dependent changes in the oligomeric state of centromere-specific nucleosomes[40]. By comparison, the unique architecture of HLp tetramers displays a high degree of rigidity, suggesting distinct functional requirements for genome organization and packaging in bacteria.

The HLp tetramer displays a continuous band of positive surface charge encircling the complex, consistent with its role in sequence-independent DNA binding. The slight variability in the binding affinity of HLp for DNA fragments with different GC content observed in EMSA and MST assays most likely results from sequence-dependent differences in intrinsic DNA properties, such as bending flexibility, helix geometry, and groove width, which in turn influence protein-DNA interactions[41–44]. Crystal structures of HLp-DNA complexes reveal the HLp tetramer to be decorated with distinct DNA-binding interfaces along its perimeter, including motifs corresponding to the canonical "paired end of helices" and "β-bridge" motifs[35]. Thereby, their spatial arrangement in HLp resembles that of eukaryotic and archaeal histones.

While the two individual DNA-bound crystal structures might hint at a DNA-bridging mode, their structural superposition and the location of the individual DNA-binding motifs suggested a potential DNA-wrapping mode for HLp. Between the two binding modes, our MD simulations indicate that the wrapping conformation exhibits a more convergent and localized conformational ensemble, consistent with enhanced thermostability. Notably, even simulations initiated from the bridging conformation spontaneously transitioned toward wrapping, yielding a final state in which ~60 bp of DNA encircle the tetramer, consistent with structural models. Nevertheless, we cannot rule out that HLp also bridges DNA in vivo, potentially in concert with other cellular factors.

Biochemical assays further support the wrapping-based binding mode. In MNase digestion assays, HLp protects DNA from nucleolytic cleavage, producing irregularly spaced fragments (35–72 bp) distinct from the regular 30-bp pattern generated by HMfB and the ~147-bp protection seen in eukaryotic nucleosomes. These fragment sizes correspond approximately to half and full turns of DNA around the HLp tetramer, indicating protection via wrapping but with flexible nucleosome-like spacing. Additional in vitro assays, including TPM, topoisomerase relaxation, and ladder EMSA, demonstrate that HLp compacts DNA and alters its topology, albeit less efficiently than HMfB. HLp does not promote the formation of circular DNA monomers in the circularization assay, suggesting that it does not bring DNA ends into close proximity for ring closure under the tested conditions.

Expression of HLp in *E. coli* leads to dramatic reorganization of the nucleoid. DAPI staining reveals marked nucleoid decondensation and increased cellular length, indicative of disrupted DNA compaction and/or interference with cell division. These phenotypes closely mirror those observed upon HMfB expression, although the nucleoid appears more homogeneous with HMfB than with HLp, suggesting differences in chromatin architecture. Consistent with these observations, MNase digestion of genomic DNA from HLp-expressing *E. coli* produced a ladder of protected DNA fragments, directly demonstrating in vivo DNA protection by HLp. The irregular banding pattern differs from the regular 30-bp ladder generated by HMfB in comparable *E. coli* chromatinization experiments[45], but closely matches the pattern observed

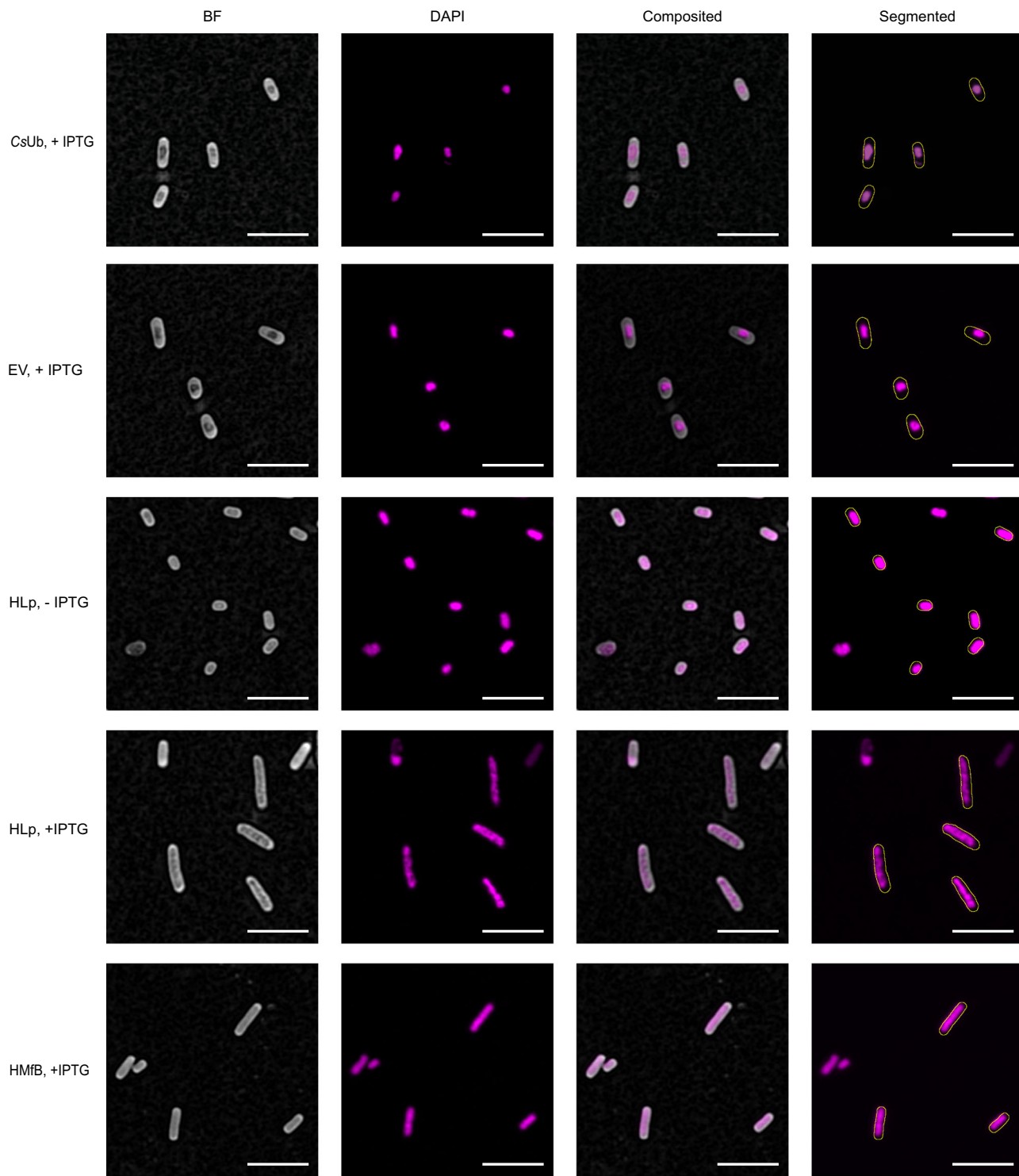

**Fig. 7 | HLp binds to genomic DNA in vivo.** Airyscan imaging of *E. coli* cells expressing HLp and HMfB (positive control) with DAPI dye, compared to negative controls transformed with the empty vector (EV), the vector expressing *Cs*Ub, and the uninduced HLp sample (-IPTG). The left column presents the brightfield image with the subtracted background. The second column displays the maximum intensity projection of fluorescence. The third one is the overlap, and the last column shows the segmented bacteria outline on top of the fluorescence channel. Scale bar: 5 μm. All experiments were performed three times with independently grown bacterial cultures, yielding similar results. Source data are provided as a Source Data file.

in our in vitro MNase digestion assay using the 600-bp-GC40 DNA fragment.

These in vivo findings support the hypothesis that HLp functions as a global chromatin organizer, similar to its homolog in *L. interrogans*, where the FtF histone is among the most highly expressed and essential proteins[23]. Notably, neither *L. interrogans* nor *L. perolatii*

encodes other histone homologs or canonical bacterial NAPs, further indicating that HLp serves as the principal chromatin component in these species. In contrast, species such as *B. bacteriovorus* encode two distinct histones (HBb and Bd3044) as well as NAPs such as HU, underscoring lineage-specific diversity in bacterial chromatin organization strategies[23,27]. Consistent with HLp's proposed central role,

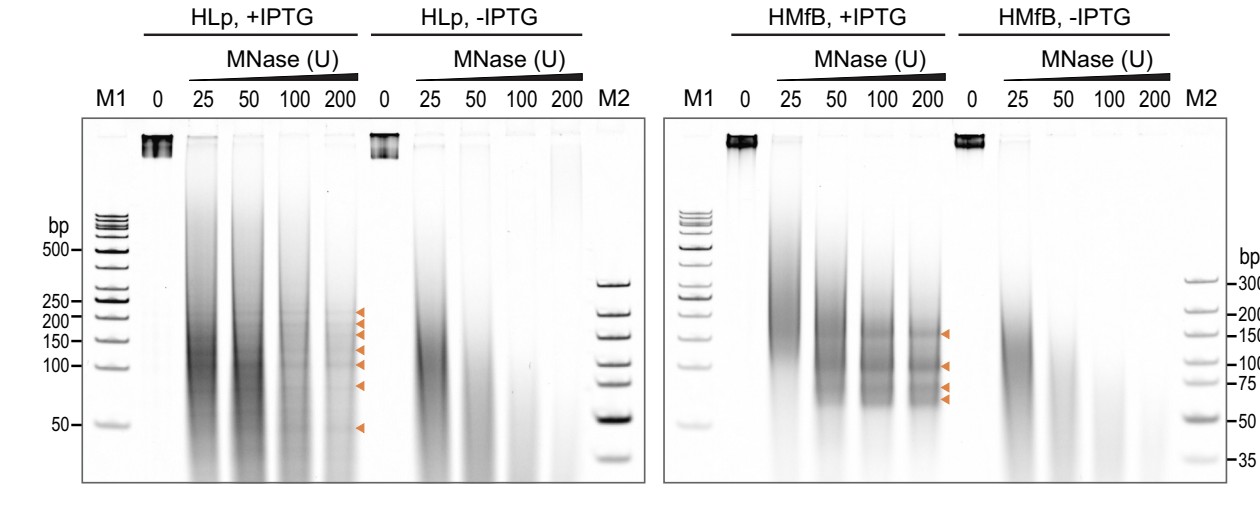

**Fig. 8 | HLp chromatinizes *E. coli* genomic DNA. a** TBE gels showing DNA fragments protected from MNase digestion in *E. coli* strains expressing HLp or HMfB, across increasing amounts of MNase. Uninduced samples (-IPTG) were included as negative controls. Each experiment was performed independently three times, yielding similar results. **b** Table summarizing the sizes of DNA fragments obtained after treatment with the highest amount of MNase. Fragment sizes were determined by densitometric analysis, using the molecular weight marker M2 (GeneRuler Ultra Low Range DNA Ladder, ThermoFisher Scientific) as the reference. The densitometric analysis was performed for the indicated number (n) of independent experiments, and the average DNA fragment lengths with standard deviations (SD) are listed. Source data are provided as a Source Data file.

| Selected bands, from top to bottom | HLp Average length ± SD (bp, n = 3) | HMfB Average length ± SD (bp, n = 3) |
|---|---|---|
| Band 1 | 204 ± 3 | 156 ± 4 |
| Band 2 | 174 ± 3 | 97 ± 3 |
| Band 3 | 151 ± 6 | 69 ± 2 |
| Band 4 | 124 ± 2 | 62 ± 2 |
| Band 5 | 98 ± 2 | - |
| Band 6 | 71 ± 1 | - |
| Band 7 | 46 ± 0 | - |

genomic analysis reveals that in *L. perolatii*, the hlp gene resides within a conserved locus adjacent to genes encoding SMC, MAP, and a putative transmembrane protein (Supplementary Fig. 1). This synteny is preserved in *L. interrogans* and other Leptospira species, suggesting a functionally co-evolved genomic module. Such an arrangement raises the possibility that HLp cooperates closely with SMC to maintain proper chromosome structure and segregation. The neighboring MAP enzyme, responsible for cleaving the initiating methionine from nascent proteins, may support efficient maturation of HLp, SMC, and other critical factors involved in nucleoid dynamics.

From an evolutionary perspective, HLp is only the second bacterial histone characterized structurally and functionally, after the dimeric histone HBb. While bacterial dimeric histones like HBb appear restricted to bacteria and nucleosomal histones are found exclusively in archaea and eukaryotes, FtF histones uniquely span both archaeal and bacterial domains. Our bioinformatic analysis demonstrates that bacterial and archaeal FtF histones form distinct and clearly separated clusters, supporting the hypothesis that FtF histones may trace their ancestry back to the last universal common ancestor (LUCA). Alternatively, the broad distribution of FtF histones across archaeal phyla, including Asgard archaea, and their sparse, patchy occurrence in bacteria, suggest a scenario involving multiple independent horizontal gene transfer events from archaea into specific bacterial lineages[13]. Thus, the evolutionary history of FtF histones likely reflects a combination of ancestral origin, lineage-specific adaptations, and recurrent gene-transfer events.

In summary, our work expands the known diversity of bacterial histones and establishes HLp as a DNA-wrapping histone that assembles into a stable homotetramer. Rather than being simplified ancestors of eukaryotic histones, prokaryotic histones represent structurally diverse, functionally adaptable proteins that contribute to genome organization through distinct DNA-binding modes. Beyond structural and mechanistic insight, future studies will need to explore how these histones function in their native cellular contexts—how they interact with other histone variants, cooperate with or substitute for NAPs, and participate in processes such as transcription, replication, and genome repair.

## Methods

### Bioinformatic analyses

For cluster analysis, we used the histone sequences HMfB from *M. fervidus* (UniProtKB ID: P19267), HBb from *B. bacteriovorus* (Q6MRM1), the *A. aeolicus* pseudodimeric histone (O66665), the FtF histone from *L. interrogans* (Q8F3E8), and HTkC from *T. kodakarensis* (Q5JDW7) as queries in BLAST searches to retrieve archaeal and bacterial histone homologs. Searches were performed via the NCBI BLAST webserver with the 'Max target sequences' parameter set to 5000[46,47]. Sequences shorter than 50 residues, longer than 200, or annotated as 'partial' or 'fragment' were excluded to eliminate truncated entries and multi-domain proteins. The full-length hits were pooled and filtered using MMseqs2 to reduce redundancy at 80% pairwise sequence identity over 80% length coverage[48]. The resulting non-redundant dataset of

3341 sequences was clustered using CLANS based on all-against-all pairwise sequence similarities computed via BLAST[49,50]. Clustering was performed to equilibrium using an E-value cut-off of 1e$^{-6}$. Genome neighborhood analysis was carried out using the EFI Genome Neighborhood Tool[51], and structural predictions were generated using AlphaFold2 (AF2)[31].

## Bacterial strains and cultivation
Cloning and plasmid amplification were conducted using *Escherichia coli* Top10. *E. coli* BL21(DE3) and *E. coli* Mutant56(DE3) were used as hosts for protein expression[52]. Bacteria were grown in LB media supplemented with appropriate antibiotics.

## Cloning, plasmids, and synthetic DNA
The nucleotide sequences of the HLp-encoding gene CH373_10155 [GenBank: NPDY01000005.1, 141608–141799 (+)] from *L. perolatii strain FH1-B-C1* and the hmfB gene (GenBank: M34778.1) were codon-optimized and synthesized (BioCat GmbH) for protein expression in *E. coli* (Supplementary Fig. 1).

HLp was expressed fused to an N-terminal (histidine)$_6$ tag using the vector pET-30a(+), and HMfB was expressed using pET-28a(+). Ubiquitin from *Caldiarchaeum subterraneum* (*Cs*Ub), used as a control for light microscopy imaging, was expressed from pET-30b(+)[37].

Synthetic oligonucleotides (Merck) used in this study are listed in Supplementary Table 1. For DNA binding analyses, complementary single-stranded (ss) oligonucleotides were mixed, heated to 95 °C for 5 min, and slowly cooled to room temperature to generate double-stranded (ds) DNA fragments. The 600-bp-GC40 and 240-bp-GC40 DNA fragments used in DNA binding assays were amplified by PCR using the vector pETHis1a as the template[53].

## Protein expression and purification
HMfB and HLp were expressed in *E. coli* Mutant56(DE3) and *E. coli* BL21(DE3), respectively. Bacterial cultures were maintained in LB broth supplemented with kanamycin at 37 °C. At an optical density of $OD_{600} = 0.6$, isopropyl-β-d-thiogalactoside (IPTG) was added at a final concentration of 1 mM to induce protein expression. Cells were agitated at 37 °C for an additional 4 h. Cells were pelleted and resuspended in buffer containing 20 mM Tris, pH 8.0, 300 mM NaCl, and 10 mM imidazole, supplemented with protease inhibitor mix (cOmplete™, EDTA-free Protease Inhibitor Cocktail, Roche), 0.1 mM PMSF, 3 mM MgCl$_2$ and DNase I (AppliChem GmbH). Cells were lysed using a French press, and cell debris and insoluble material were pelleted by centrifugation at 95,000 x *g* for 45 min. The supernatant was filtered through a 0.45 µm filter and applied to a 5 mL HisTrap column (Cytiva). Bound proteins were eluted using a linear imidazole gradient, ranging from 10–500 mM, in the aforementioned buffer. For HLp, the (histidine)$_6$ tag was cleaved with *Tobacco Etch Virus* (TEV) protease. A second purification step using a HisTrap column was performed using the aforementioned buffers containing 150 mM NaCl to separate the protein from the cleaved tag. Purified proteins were dialyzed against 20 mM Tris buffer at the indicated pH with 150 mM NaCl and concentrated using an Amicon Ultra Centrifugal Filter Device (3 kDa MWCO, Millipore). Protein purity was confirmed by SDS-PAGE (mPAGE 4–12% Bis-Tris Precast Gel, Millipore), and protein concentration was determined spectrophotometrically or using a BCA protein assay (ThermoFisher Scientific). For the tethered particle motion (TPM) assay, purified HLp was dialyzed against 50 mM Tris, pH 7.0, 75 mM KCl, and 10% glycerol.

## Circular dichroism (CD) spectroscopy
HLp was dialyzed against 10 mM phosphate buffer, pH 8.0, 75 mM KF, and diluted to a final concentration of 15 µM. CD spectra were recorded on a Jasco J-810 spectrometer (JASCO) in the wavelength range of 190–250 nm using a cuvette with a 1 mm path length and a 100 nm/min reading speed. A total of ten single spectra were recorded and averaged. To analyze the thermal stability of HLp, the ellipticity (θ) was measured at 222 nm over a temperature gradient of 10–100 °C, using a ramp of 1 °C/min, a data pitch of 0.5, and a response time of 1 s. Data analysis, blank subtraction, and curve smoothing were performed using the Spectra Manager software 1.53 (JASCO). CD spectra and thermal melting curves were plotted using GraphPad Prism 9 (GraphPad Software, Inc.).

## Size exclusion chromatography coupled with multi-angle light scattering (SEC-MALS)
The HLp-DNA complex was analyzed by SEC-MALS following incubation of HLp with the 30-bp-GC40 DNA fragment at a protein-to-DNA molar ratio of 2:1. All samples were incubated at 37 °C for 10 min, and aggregates were removed by centrifugation. Samples were applied onto a Superdex 75 Increase 10/300 GL column (Cytiva), pre-equilibrated with SEC buffer (25 mM Tris, 50 mM NaCl, 50 mM KCl, pH 7.5). The run was performed at a 0.5 mL/min flow rate on a 1260 Infinity II HPLC system (Agilent) coupled to a miniDAWN TREOS and Optilab T-rEX refractive index detector (Wyatt Technology). HLp and the 30-bp-GC40 DNA fragment were detected at 215 nm and 260 nm, respectively. Measurements were performed in triplicate, and molecular mass distributions were calculated using the ASTRA v.7.3.0.18 software suite (Wyatt Technology).

## Crystallization, data collection, and structure determination
Crystal structures of free HLp and two HLp-DNA complexes were solved by molecular replacement and refined using standard crystallographic procedures. Structures were deposited in the PDB under accession codes 9QT0 (free HLp), 9QT1 (HLp-DNA_1), and 9QT2 (HLp-DNA_2). Crystallization conditions, data processing, and refinement details are provided in the Supplementary Methods and Supplementary Table 3.

## Molecular dynamics (MD) simulation
All-atom molecular dynamics (MD) simulations were carried out in GROMACS (version 2023.2) to evaluate the stability of HLp-DNA complexes in both wrapping and bridging configurations[54]. Initial models were constructed from HLp-DNA crystal structures and refined by symmetry expansion, trimming, and reconnection of DNA fragments. Each system was equilibrated and subjected to multi-microsecond simulations, followed by residue-wise contact, hydrogen bond, and free-energy landscape analyses (Fig. 5 and Supplementary Figs. 9–19). Full simulation protocols and analysis procedures are provided in the Supplementary Methods.

## DNA binding assays
All DNA-binding assays were performed following established protocols with minor modifications[27]. We employed a set of complementary in vitro and in vivo approaches, including EMSA, micrococcal nuclease (MNase) digestion, tethered particle motion (TPM), DNA topology, ligase-mediated circularization, and microscale thermophoresis (MST), with HMfB included for comparison. Full experimental protocols and analysis details are provided in the Supplementary Methods. Uncropped and unprocessed gel scans are provided in the Source Data File.

## Light microscopy imaging and data processing
The following *E. coli* cultures were used: *E. coli* BL21(DE3) transformed with the empty vector pET-30a(+) or with pET-30a(+) encoding *C. subterraneum* ubiquitin (*Cs*Ub) (both serving as negative controls), BL21(DE3) transformed with pET-30a(+) encoding HLp, and *E. coli* Mutant56(DE3) transformed with pET-28a(+) expressing HMfb. Microscopic analyses were conducted on three independent replicates of each strain, each derived from separate colonies.

**Sample preparation.** Overnight cultures of each strain were diluted 30-fold (V/V) in LB medium supplemented with the appropriate antibiotic and incubated at 37 °C, 170 rpm. Except for the negative controls, protein expression was induced with IPTG at a final concentration of 1 mM at an $OD_{600} = 0.4$–0.6. Following cultivation for another 4 h, 500 µL bacterial culture was pelleted. The cells were washed with PBS and fixed using 4% (V/V) formaldehyde. Fixation was stopped with glycine at a final concentration of 150 mM. Following two wash steps with PBS, the cells were incubated with 1 µg/mL DAPI solution (Invitrogen), washed, and resuspended in PBS. The treated cells were dropped onto a 2% agarose pad, which was placed upside down in a 35 mm imaging dish (High Glass Bottom, Ibidi) for imaging. Triplicates of each sample were analyzed.

**Imaging and image processing.** Light microscopy imaging was conducted using a Zeiss LSM 780 inverted confocal microscope (Carl Zeiss AG, Oberkochen, Germany) equipped with a 63 x oil/1.4NA oil-immersion objective. The 405 nm diode laser line was used for DAPI excitation and brightfield imaging. Airyscan datasets were processed with Airyscan software to generate 32-bit images using pixel reassignment and Wiener filter-based deconvolution[55]. Brightfield images were used to segment individual bacterial cells via a custom-developed Macro in FIJI (version 2.3.0), which included background subtraction and auto-thresholding[56]. Segmentation results were manually curated to eliminate artefacts. For fluorescence analysis, a maximum intensity projection was first applied to the Airyscan z-stacks. The segmented bacterial outlines were overlaid onto the fluorescence channel, followed by a second segmentation step on the fluorescence channel to identify nucleoid regions using auto-thresholding.

### Reporting summary
Further information on research design is available in the Nature Portfolio Reporting Summary linked to this article.

## Data availability
Coordinates and structure factors of the crystal structures have been deposited in the PDB under entry numbers 9QT0 (DNA-free HLp; https://doi.org/10.2210/pdb9QT0/pdb), 9QT1 (HLp-DNA_1; https://doi.org/10.2210/pdb9QT1/pdb), and 9QT2 (HLp-DNA_2; https://doi.org/10.2210/pdb9QT2/pdb)[57–59]. For the MD analysis, all configuration files, trajectory files, movies, and analysis scripts are deposited at Zenodo (https://doi.org/10.5281/zenodo.15234989)[60]. The TPM dataset has been deposited in the 4TU.ResearchData repository (https://data.4tu.nl) and is accessible via the https://doi.org/10.4121/5b604dd5-5498-46aa-b437-775e957f93e3 (https://doi.org/10.4121/5B604DD5-5498-46AA-B437-775E957F93E3.V2)[61]. All remaining raw data are available in the Source Data file provided with this paper. Requests for materials should be addressed to the corresponding authors. Source data are provided in this paper.

## Code availability
Code used to determine the angle between non-contacting helices has been provided as Supplementary Code.

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

## Acknowledgements

We thank the staff of Beamline X10SA of the Swiss Light Source (PSI, Villigen, Switzerland) for excellent technical support. We are grateful to Reinhard Albrecht for assistance with crystallization and crystallographic data collection. We extend our thanks to Linxuan Li (Dept. of Integrative Evolutionary Biology, MPI for Biology Tübingen, Germany) and Agnes Henschen (Dept. of Algal Development and Evolution, MPI for Biology Tübingen, Germany) for their support with light microscopy sample preparation. We thank Pedro Escudeiro (Dept. of Protein Evolution, MPI for Biology Tübingen, Germany) for proofreading the manuscript. We acknowledge the HPC system Raven at the Max Planck Computing and Data Facility for the performed computational work. This work was supported by institutional funds from the Max Planck Society (Y.H., K.Q., K.B., H.R., A.P., A.N.L., M.D.H., V.A., and B.H.A.) and by funding from the Netherlands Organization for Scientific Research (OCENW.GROOT.2019.012) for S.S. and R.T.D.

## Author contributions

Conceived and designed the experiments: A.N.L., B.H.A., V.A., and Y.H. Performed the experiments: A.P., H.R., K.B., S.S., and Y.H. Performed MD simulations: K.Q. and Y.Z. Performed bioinformatic analyses: V.A. Analyzed the data: A.P., B.H.A., M.D.H., S.S., R.T.D., V.A., and Y.H. Wrote the paper: B.H.A., V.A., Y.H., and K.Q., with contributions from the other authors.

## Funding

## Competing interests

The authors declare no competing interests.
