## [Transparent Peer Review file · Nature Communications]

DNA Wrapping by a Tetrameric Bacterial Histone

Corresponding Author: Dr Birte Hernandez Alvarez

Version 0:

Reviewer comments:

Reviewer #1

(Remarks to the Author)

The manuscript by Hu et al. uses a large variety of experimental approaches complemented by in silico approaches characterize the bacterial histone protein HLP, following a similar approach by the same research team earlier for a related bacterial histone. Overall, this is a well-carried out study and a well written manuscript. Given the emerging role of bacterial histone proteins the manuscript is timely and likely of large interest for the readership of Nature Communications. I have only a few very minor concerns the authors should address

Minor Points:

- Given the availability of AF-multimer it remained unclear if the authors used a structure of AF-generated monomer or tetramer to solve the crystal structure
- Regarding the stronger binding observed for the GC-rich DNA in the MST experiments, it would be good if the authors would be more clear in the main text that it is only a minimal increase of about ~2-fold based on the Kd numbers provided in the supplementary material
- EMSA-experiment shown in Figure S5. Even though the 40%GC shows clearly the strongest interaction can the authors think of an explanation that with 50% binding get reduced whereas at 60% it starts to get stronger again?
- As it is not a common E. coli strain, the authors should describe the mutant56 strain or at least provide a reference.
- For some figures the choice of colors and general representation might be reconsidered to increase visibility:
 - o Fig. 1: Panel A might benefit from more contrast rich color choices to distinguish the different classifications easier.
 - o Fig. S1: Panel A is currently using a too small font to be readable.
 - o Fig. S3: Panel B the scale for the electrostatics is currently not provided.
 - o Fig. S4: Figure quite dark and individual dimers hard to distinguish.

Reviewer #2

(Remarks to the Author)

Summary:

The authors describe the characterisation of a putative histone protein in the bacterial pathogen *Leptospira perolatii*. After a clearly written and scholarly introduction to the burgeoning array of prokaryote histones they perform a bioinformatic sequence analysis which nicely complements and extends their previous work on *Bdellovibrio bacteriovorus* (<https://doi.org/10.1038/s41467-024-52337-y>) to identify an experimentally amenable and novel Face-to-Face $\alpha 3$ histone in *L. perolatii* ("HLP").

An artificial expression system is constructed for HLP in *E. coli*, and purified protein is characterised using a wide variety of

methods, often in direct comparison with the well-studied histone HMfB from the archaeon *Methanothermus fervidus*: Size exclusion/light scattering suggests a tetrameric form for HLP, and a homotetramer with characteristic histone folds and distinct oligomerisation interface is revealed in high-resolution X-ray crystal structure; EMSA, microscale thermophoresis and SEC-MALS indicate DNA binding and this is confirmed by identification of two X-ray crystal structures; the distribution of charged residues in HLP suggests that DNA could wrap the tetramer, but this possibility is not entirely confirmed by the DNA:protein X-ray structures – the authors therefore apply a battery of in vitro/biophysics assays and a detailed MD simulation which all suggest changes in DNA association and topology consistent with a more nucleosome-like (yet distinct) DNA wrapping.

To assess in vivo genome binding function, the authors turn to their *E. coli* expression system and partially recapitulate observations made by the Warnecke lab (<https://doi.org/10.7554/eLife.49038>) in which expression of prokaryote histone alters gross nucleoid shape and cell length. The authors conclude that: “that HLP assembles into a unique, DNA-wrapped tetramer, expanding the known repertoire of bacterial histone architectures,”

Opinion:

The field of non-eukaryote/non-standard “chromatin” is hot at the moment and should be of interest to a wide audience. The in vitro, and especially the crystallographic, evidence presented here seems compelling – although see my minor comments below. Enough is presented in the Materials and Methods to ensure reproducibility, and code and PDB accessions are available. My main issue is with the in vivo evidence, which I feel needs to be stronger to completely nail the important observation that HLP is binding the nucleoid as a NAP/“chromatin”. I appreciate that the authors may not have the containment or culture facilities to grow *Leptospira*, but some evidence of DNA wrapped HLP in vivo is required in my view. The Warnecke Lab “chromatinization” of *E. coli* experiments not only assayed cells microscopically but used in vivo MNase digestion to reveal histone protected DNA regions. I’m not suggesting that the authors attempt a full MNase-seq experiment, but they could certainly test for MNase-resistant ladder creation (<https://doi.org/10.7554/eLife.49038.001> Figure 1) during expression of HLP in the *E. coli* host.

Minor technical comments:

Line 342. The MNase buffer is just Tris and NaCl. I’m amazed that this digestion works without any added Mg²⁺ and/or Ca²⁺

EMSA. I grew up with EMSAs in which DNA was labelled, and reactions contained some non-specific competitor such as poly dI.dC. These reactions were then challenged with various unlabelled competitor DNAs to measure specificity – many slightly charged proteins will bind any DNA fragment in a simple DNA-stain EMSA like the ones used here. That said, the authors have explored the in vitro DNA binding of HLP with an excellent battery of tools, but I think they should be careful with the “specificity” wording associated with their EMSA assays.

Reviewer #3

(Remarks to the Author)

The existence of “histones” in bacteria became an uprising topic after the efficient protein structure breakthrough tool of AlphaFold. In this work, following several previous works by the same authors, Hu et al. investigated the structural and functional properties of HLP, a bacterial histone from *Leptospira perolatii*, through crystallography, biochemical assays, and atomistic molecular dynamics (MD) simulations. The authors showed that HLP forms a stable homotetramer and binds DNA via a wrapping mechanism that encloses approximately 60 base pairs. Crystal structures revealed multiple protein–DNA interaction motifs, and MD simulations further demonstrated that the wrapping mode is thermodynamically more stable than a bridging configuration, with spontaneous transitions observed during simulation. These results are supported by in vitro assays showing HLP-induced DNA compaction and topology modulation. Previous studies primarily focused on bacterial histones like HBB, which bind DNA through bending, but HLP is the first to demonstrate a DNA-wrapping mode similar to that of archaeal FtF histones. The integration of MD simulation adds a mechanistic understanding of HLP-DNA dynamics, which strengthens the structural findings. The work provides new insights into the structural diversity of bacterial chromatin organization and may be considered for publication after addressing the points below.

Major issues:

- 1) Tetramer formation among histones is not uncommon, and several related studies have been reported in the field. The manuscript would benefit from a more thorough engagement with relevant prior literature. Several important studies on histone tetramer formation are not cited here. For example, in addition to the Archaeal histone tetramer studies that this work cited, Black et al. demonstrated an in vitro reconstituted CENP-A/H4 tetramer (Black et al., *Nature*, 2004), while Dalal et al. described the tetrameric structure of CENP-A-containing nucleosomes in interphase *Drosophila* cells (Dalal et al., *PLoS Biology*, 2007).
- 2) It seems that the HLP protein tetramer remained so planar throughout the 1-microsecond simulation, which is somewhat surprising and very interesting. Is that true? It would be helpful if the authors performed some simple geometry analyses, such as measuring the dihedral angle between the planes of the two dimers. If so, it shows the strong support from the two binding interfaces between the dimers. To my knowledge, this is the first reported histone tetramer structure featuring two distinct dimer-dimer binding interfaces. The author may compare it with the traditional tetrameric interface, the so-called four-helix bundle to highlight the uniqueness of current structure. Previously, Winogradoff et al. reported shearing motion at the histone dimer:dimer interface in molecular dynamics simulations of the nucleosome (Winogradoff et al. *Scientific Reports*, 2015). Zhao et al. examined DNA orientation around the traditional histone tetramer (Zhao et al., *Biophysical Journal*, 2019). It is possible that the planar placement of DNA fits the relatively low demand of compacting nucleoid DNAs in bacteria. These related studies can be discussed to place the current findings in a broader context.
- 3) Figure 5B presents the time evolution of RMSD and protein–DNA contact numbers during the MD simulations, the authors

mention that only one representative trajectory was analyzed for each model. However, it is unclear how the error bars were calculated for either the RMSD of the DNA backbone or the contact counts. Clarification on this point would be appreciated.

4) In line 621, the authors state that “the potential energy of the wrapping model is lower than the bridging model,” referring to Figure S7A, to support the claim that the wrapping mode offers greater thermodynamic stability. However, I would be cautious in drawing thermodynamic conclusions solely from potential energy comparisons in MD simulations, especially when the initial configurations and DNA lengths differ between models. Potential energy of MD simulation is highly dependent on the specific setup and starting conformations, and in this case, the wrapping and bridging models were constructed with different initial energies. A direct comparison of absolute potential energies between two distinct systems may not be meaningful. In fact, if both models were normalized to an identical starting potential energy level (e.g., -500,000), it might even appear that the bridging model would show a lower final potential energy, potentially reversing the conclusion.

5) Moreover, Figure S7B&C indicate that the bridging model has a slightly higher number of protein–DNA contacts, and Figure 5B shows that after approximately 500 ns, the bridging model maintains more contacts than the wrapping model. These observations may suggest that the bridging configuration is at least comparably stable, if not more so, under the conditions simulated. Given these points, the argument for the wrapping model’s greater thermodynamic stability may be more convincing if supported by additional analyses, e.g. free energy estimates, replicated trajectories.

Minor Issues and Clarifications:

1) Line 609: The sentence beginning with “we generated two starting models...” was difficult to follow. While the text refers to two models—wrapping and bridging—the cited figure (Fig. S6) appears to show two different starting models for the wrapping mode only, which differs from what is shown in Fig. 5A. This inconsistency made it unclear whether both the wrapping and bridging models had two starting conformations each, or only one of them did. Clarifying this part of the text, possibly by specifying how many initial models were created and which ones were used in the final analysis, would greatly aid reader understanding.

2). Trajectory Selection: Could the authors elaborate on the rationale for choosing a single “representative” trajectory for each model in the final analysis? Given that two independent simulations were performed, it would be more robust to report averaged quantities across both trajectories, or at least provide justification for the representative selection.

3). Figure 5 Caption, Line 953: “represent” -> “representing”

4). Line 292–294 – Protein–DNA Contact Definition: The definition of a protein–DNA contact is not clearly worded. A more precise phrasing would be: “A protein–DNA contact was defined and counted if any protein heavy atom was within 0.4 nm of a DNA heavy atom.” Rewriting this sentence for clarity would make the methodology more accessible, especially to readers less familiar with contact analysis in MD simulations.

Reviewer #4

(Remarks to the Author)

Version 1:

Reviewer comments:

Reviewer #1

(Remarks to the Author)

The authors have addressed all comments of the reviewers raised in the previous round in an appropriate manner. Further, the additional data provided strengthens the complete manuscript. I do not have any additional concerns and congratulate the authors to a very nice study.

Reviewer #2

(Remarks to the Author)

This is a re-review, and I'm satisfied that you have addressed my three comments.

Reviewer #3

(Remarks to the Author)

Most of my concerns have been addressed. Here are some minor issues regarding the 5th response:

1. SI figures like Fig S11 lacks of unit for the color bar.

2. According to the SI Methods section, the trajectories were aligned to a common reference structure. However, it appears that the two landscapes in Figure S11 were computed using different reference structures, as the two binding modes are initially distinct and the authors did not specify that the same reference was used. If this is the case, the principal components (PC1 and PC2) derived from the two binding modes are not directly comparable, as they represent different coordinate bases. Therefore, comparing these two free energy landscapes side by side may not be meaningful.

To provide a more interpretable comparison, I would suggest projecting the trajectories onto biophysically meaningful parameters—such as the number of contacts between the DNA and the protein, the RMSD, or the radius of gyration (Rg) of the DNA—some used in the main text. It would allow for a more consistent and physically interpretable comparison between the two binding modes.

Reviewer #4

(Remarks to the Author)

In the following response to the reviewers' comments, our replies are in green.

REVIEWER COMMENTS

Reviewer #1 (Remarks to the Author):

The manuscript by Hu et al. uses a large variety of experimental approaches complemented by in silico approaches characterize the bacterial histone protein HLP, following a similar approach by the same research team earlier for a related bacterial histone. Overall, this is a well-carried out study and a well written manuscript. Given the emerging role of bacterial histone proteins the manuscript is timely and likely of large interest for the readership of Nature Communications. I have only a few very minor concerns the authors should address

Minor Points:

- Given the availability of AF-multimer it remained unclear if the authors used a structure of AF-generated monomer or tetramer to solve the crystal structure.

As described in the *Methods* section, we performed molecular replacement using the AF-generated dimer as a search model to solve the crystal structure of HLP. For improved clarity, this information has been also added to the main text.

- Regarding the stronger binding observed for the GC-rich DNA in the MST experiments, it would be good if the authors would be more clear in the main text that it is only a minimal increase of about ~2-fold based on the Kd numbers provided in the supplementary material

The main text has been updated to include this information in the appropriate manner.

- EMSA-experiment shown in Figure S5. Even though the 40%GC shows clearly the strongest interaction can the authors think of an explanation that with 50% binding get reduced whereas at 60% it starts to get stronger again?

We thank the reviewer for this observation and have revised the main text accordingly. Histones, including HLP, primarily interact with the phosphate backbone and minor groove of DNA through electrostatic contacts. The interaction interface between HLP and DNA resembles that of eukaryotic nucleosomes and archaeal hypernucleosomes, indicating evolutionarily conserved bending and groove requirements. The variation observed in HLP's binding affinity to 30 bp DNA fragments with variable GC content ranging from 40-60% likely reflects sequence-dependent structural and biophysical properties of the DNA fragments, including bending flexibility, minor groove geometry, and electrostatic potentials. AT-rich DNA is generally more flexible with narrower

minor grooves that enhance electrostatic interactions between charged protein side chains and the DNA backbone. Conversely, GC-rich DNA is more rigid and thermodynamically stable (Rohs et al., 2009; West et al., 2010; Faltejsková et al., 2020; Zuccheri et al., 2001). Base stacking and intrinsic curvature also modulate local DNA flexibility and histone affinity (Yakovchuk et al., 2006; Freeman et al., 2014; Virstedt et al., 2004).

Our DNA fragments were designed with 10-bp repeat motifs. The 50% GC fragment alternates equal AT and GC blocks, while the 40% and 60% GC fragments vary the lengths of these regions relative to each other (see Supplementary Table S1). These local variations likely alter rigidity and flexibility, which explains the nonlinear binding pattern.

Additionally, the higher thermal stability of GC-rich DNA may reduce end fraying in annealed fragments, thereby influencing protein-DNA interactions. Furthermore, sequence-dependent mispairing during oligonucleotide annealing may form DNA multiplexes that affect the electrophoretic mobility of DNA-protein complexes in EMSA.

- As it is not a common *E. coli* strain, the authors should describe the mutant56 strain or at least provide a reference.

The according reference is now provided in the *Methods* section.

- For some figures the choice of colors and general representation might be reconsidered to increase visibility:

We thank the reviewer for this helpful suggestion.

- o Fig. 1: Panel A might benefit from more contrast rich color choices to distinguish the different classifications easier.

In the revised version of Fig. 1A, we have adjusted the color scheme to include more contrast-rich colors, which improves the visual distinction between the different classifications.

- o Fig. S1: Panel A is currently using a too small font to be readable.

To improve readability, we increased the font size in the original Supplemental Fig. 1A and separated panels A and B into two standalone figures, which now appear as Suppl. Fig. 1 and Suppl. Fig. 2.

- o Fig. S3: Panel B the scale for the electrostatics is currently not provided.

In the updated version of Fig. S3B, we have added the relevant labels to the electrostatics scale bar.

o Fig. S4: Figure quite dark and individual dimers hard to distinguish.

In the revised version of Supplementary Fig. 4, we have enhanced the contrast between the grey tones to improve the visibility of the individual dimers.

Reviewer #2 (Remarks to the Author):

Summary:

The authors describe the characterisation of a putative histone protein in the bacterial pathogen *Leptospira perolatii*. After a clearly written and scholarly introduction to the burgeoning array of prokaryote histones they perform a bioinformatic sequence analysis which nicely complements and extends their previous work on *Bdellovibrio bacteriovorus* (<https://doi.org/10.1038/s41467-024-52337-y>) to identify an experimentally amenable and novel Face-to-Face $\alpha 3$ histone in *L. perolatii* (“HLp”).

An artificial expression system is constructed for HLp in *E. coli*, and purified protein is characterised using a wide variety of methods, often in direct comparison with the well-studied histone HMfB from the archaeon *Methanothermus fervidus*: Size exclusion/light scattering suggests a tetrameric form for HLp, and a homotetramer with characteristic histone folds and distinct oligomerisation interface is revealed in high-resolution X-ray crystal structure; EMSA, microscale thermophoresis and SEC-MALS indicate DNA binding and this is confirmed by identification of two X-ray crystal structures; the distribution of charged residues in HLp suggests that DNA could wrap the tetramer, but this possibility is not entirely confirmed by the DNA:protein X-ray structures – the authors therefore apply a battery of in vitro/biophysics assays and a detailed MD simulation which all suggest changes in DNA association and topology consistent with a more nucleosome-like (yet distinct) DNA wrapping.

To assess in vivo genome binding function, the authors turn to their *E. coli* expression system and partially recapitulate observations made by the Warnecke lab (<https://doi.org/10.7554/eLife.49038>) in which expression of prokaryote histone alters gross nucleoid shape and cell length. The authors conclude that: “that HLp assembles into a unique, DNA-wrapped tetramer, expanding the known repertoire of bacterial histone architectures,”

Opinion:

The field of non-eukaryote/non-standard “chromatin” is hot at the moment and should be of interest to a wide audience. The in vitro, and especially the crystallographic,

evidence presented here seems compelling -although see my minor comments below. Enough is presented in the Materials and Methods to ensure reproducibility, and code and PDB accessions are available. My main issue is with the *in vivo* evidence, which I feel needs to be stronger to completely nail the important observation that HLP is binding the nucleoid as a NAP/"chromatin". I appreciate that the authors may not have the containment or culture facilities to grow *Leptospira*, but some evidence of DNA wrapped HLP *in vivo* is required in my view. The Warnecke Lab "chromatinization" of *E. coli* experiments not only assayed cells microscopically but used *in vivo* MNase digestion to reveal histone protected DNA regions. I'm not suggesting that the authors attempt a full MNase-seq experiment, but they could certainly test for MNase-resistant ladder creation (<https://doi.org/10.7554/eLife.49038.001> Figure 1) during expression of HLP in the *E. coli* host.

We thank the reviewer for this valuable suggestion, which we have implemented. MNase digestion of *E. coli* cells expressing HLP revealed an MNase-resistant DNA ladder, similar to that observed for HMfB. This indicates that HLP protects genomic DNA *in vivo*. Although the overall ladder pattern resembles that of HMfB, the distinct band sizes correspond to those obtained in our *in vitro* digestion of the 600-bp-GC40 fragment in the presence of HLP. We have added a corresponding figure (Figure 8), and we have revised the *Results*, *Methods*, and *Discussion* sections accordingly.

Minor technical comments:

Line 342. The MNase buffer is just Tris and NaCl. I'm amazed that this digestion works without any added Mg²⁺ and/or Ca²⁺

We apologize for this confusion. Of course, there was Ca²⁺ at a concentration of 1 mM present in the assay buffer. We have added this information to the relevant section in *Methods*.

EMSAs. I grew up with EMSAs in which DNA was labelled, and reactions contained some non-specific competitor such as poly dI.dC. These reactions were then challenged with various unlabelled competitor DNAs to measure specificity – many slightly charged proteins will bind any DNA fragment in a simple DNA-stain EMSA like the ones used here. That said, the authors have explored the *in vitro* DNA binding of HLP with an excellent battery of tools, but I think they should be careful with the "specificity" wording associated with their EMSA assays.

We thank the reviewer for this important comment. In our study, the dye was applied only for post-staining and therefore did not influence the mobility of DNA–protein complexes. We recognize that charged proteins can associate nonspecifically with

DNA in EMSAs, which often leads to diffuse smears or unstable complexes. By contrast, HLP consistently produced sharp, well-defined shifts with the 80-bp fragment, randomized 30-bp fragments of varying GC content, and with a DNA ladder. These consistent patterns support the interpretation that HLP binds DNA in a sequence-independent manner, but with reproducible, slightly different affinities, in agreement with our MST results. At the same time, we acknowledge that EMSAs without competitor DNA cannot rigorously establish sequence specificity. To avoid possible misinterpretations, we have revised the *Abstract*, *Results*, and *Discussion*, including the EMSA section, to correct the imprecise use of “specificity” and to replace the word “non-specific” with the more accurate term “sequence-independent.”

Reviewer #3 (Remarks to the Author):

The existence of “histones” in bacteria became an uprising topic after the efficient protein structure breakthrough tool of AlphaFold. In this work, following several previous works by the same authors, Hu et al. investigated the structural and functional properties of HLP, a bacterial histone from *Leptospira perolatii*, through crystallography, biochemical assays, and atomistic molecular dynamics (MD) simulations. The authors showed that HLP forms a stable homotetramer and binds DNA via a wrapping mechanism that encloses approximately 60 base pairs. Crystal structures revealed multiple protein–DNA interaction motifs, and MD simulations further demonstrated that the wrapping mode is thermodynamically more stable than a bridging configuration, with spontaneous transitions observed during simulation. These results are supported by in vitro assays showing HLP-induced DNA compaction and topology modulation. Previous studies primarily focused on bacterial histones like HBB, which bind DNA through bending, but HLP is the first to demonstrate a DNA-wrapping mode similar to that of archaeal FtF histones. The integration of MD simulation adds a mechanistic understanding of HLP-DNA dynamics, which strengthens the structural findings. The work provides new insights into the structural diversity of bacterial chromatin organization and may be considered for publication after addressing the points below.

Major issues:

- 1) Tetramer formation among histones is not uncommon, and several related studies have been reported in the field. The manuscript would benefit from a more thorough engagement with relevant prior literature. Several important studies on histone tetramer formation are not cited here. For example, in addition to the Archaeal histone tetramer studies that this work cited, Black et al. demonstrated an in vitro reconstituted CENP-

A/H4 tetramer (Black et al., Nature, 2004), while Dalal et al. described the tetrameric structure of CENP-A-containing nucleosomes in interphase *Drosophila* cells (Dalal et al., PLoS Biology, 2007).

We thank the reviewer for this important comment and fully acknowledge that tetramer formation has been described for eukaryotic and archaeal histones in several prior studies, including the citations mentioned by the reviewer. In the revised manuscript, we have expanded the introduction to provide a more comprehensive overview of different types of histone tetramers, including (H3–H4)₂ and (CENP-A-H4)₂ heterotetramers, archaeal nucleosomal tetramer formation upon DNA binding, and archaeal FtF homotetramers.

2) It seems that the HLP protein tetramer remained so planar throughout the 1-microsecond simulation, which is somewhat surprising and very interesting. Is that true? It would be helpful if the authors performed some simple geometry analyses, such as measuring the dihedral angle between the planes of the two dimers. If so, it shows the strong support from the two binding interfaces between the dimers. To my knowledge, this is the first reported histone tetramer structure featuring two distinct dimer-dimer binding interfaces. The author may compare it with the traditional tetrameric interface, the so-called four-helix bundle to highlight the uniqueness of current structure. Previously, Winogradoff et al. reported shearing motion at the histone dimer:dimer interface in molecular dynamics simulations of the nucleosome (Winogradoff et al. Scientific Reports, 2015). Zhao et al. examined DNA orientation around the traditional histone tetramer (Zhao et al., Biophysical Journal, 2019). It is possible that the planar placement of DNA fits the relatively low demand of compacting nucleoid DNAs in bacteria. These related studies can be discussed to place the current findings in a broader context.

We thank the reviewer for this thoughtful comment. As suggested, we examined the geometry of the HLP tetramer during our MD simulations. To quantify planarity, we defined the two dimers (residues 1–116 and 117–232) and determined the best-fit plane of each dimer using the C α atoms via singular value decomposition. The normal vectors of these planes were then extracted, and the inter-plane angle was calculated from their dot product for each frame of the trajectory. As shown in Supplementary Fig. 12, the dihedral angle between the two dimers in both the wrapping and bridging models remained remarkably stable, with fluctuations of less than 10°. This demonstrates that the HLP tetramer preserves a planar architecture throughout the 1- μ s simulations.

This stability contrasts with canonical nucleosomal histone tetramers, which are stabilized by a four-helix bundle and are known to exhibit significant plasticity

(Winogradoff et al., 2015; Zhao et al., 2019). We have emphasized this point in the revised *Discussion*, highlighting HLP as a histone tetramer with two distinct dimer-dimer interfaces that together enforce a rigid and planar geometry. We also place these findings in a broader biological context, proposing that the architecture of HLP tetramers represents an alternative tetramer-based mechanism of DNA wrapping, potentially adapted to the relatively modest demands of nucleoid compaction in bacteria, possibly in cooperation with other nucleoid-associated proteins.

3) Figure 5B presents the time evolution of RMSD and protein–DNA contact numbers during the MD simulations, the authors mention that only one representative trajectory was analyzed for each model. However, it is unclear how the error bars were calculated for either the RMSD of the DNA backbone or the contact counts. Clarification on this point would be appreciated.

We thank the reviewer for raising this point. For Figure 5b, only one representative trajectory was analyzed for each model; the corresponding plots for the additional trajectories are now visualized in Supplementary Fig. 10. To reduce high-frequency noise, we applied a moving-average window of 100 frames to the RMSD and protein–DNA contact number traces. Within the same window, we calculated the standard deviation, and the shaded regions (or error bars) therefore represent ± 1 standard deviation around the moving average. These values capture the local fluctuations within the trajectory rather than variability across independent simulations. For transparency, all analysis scripts are available in our Zenodo repository (<https://zenodo.org/records/15234989>).

4) In line 621, the authors state that “the potential energy of the wrapping model is lower than the bridging model,” referring to Figure S7A, to support the claim that the wrapping mode offers greater thermodynamic stability. However, I would be caution in drawing thermodynamic conclusions solely from potential energy comparisons in MD simulations, especially when the initial configurations and DNA lengths differ between models. Potential energy of MD simulation is highly dependent on the specific setup and starting conformations, and in this case, the wrapping and bridging models were constructed with different initial energies. A direct comparison of absolute potential energies between two distinct systems may not be meaningful. In fact, if both models were normalized to an identical starting potential energy level (e.g., -500,000), it might even appear that the bridging model would show a lower final potential energy, potentially reversing the conclusion.

We thank the reviewer for this insightful comment. We agree that absolute potential energies in MD simulations are highly dependent on setup parameters (force field, initial conformations, system size/DNA length, etc.) and therefore cannot be used as a direct measure of thermodynamic stability when comparing distinct models such as wrapping and bridging. Likewise, protein–DNA contact numbers are best interpreted as indicators of binding geometry rather than as proxies for thermodynamic stability. Our intention was not to draw firm thermodynamic conclusions from these analyses. To prevent over-interpretation, we have removed Supplementary Fig. S7A and revised the text to tone down stability-related statements.

5) Moreover, Figure S7B&C indicate that the bridging model has a slightly higher number of protein–DNA contacts, and Figure 5B shows that after approximately 500 ns, the bridging model maintains more contacts than the wrapping model. These observations may suggest that the bridging configuration is at least comparably stable, if not more so, under the conditions simulated. Given these points, the argument for the wrapping model’s greater thermodynamic stability may be more convincing if supported by additional analyses, e.g. free energy estimates, replicated trajectories.

We thank the reviewer for this constructive comment. Following the suggestion, we performed an additional free-energy landscape (FEL) analysis for both models, projecting the trajectories onto collective variables defined by protein C α and DNA phosphate atoms. As shown in Supplementary Fig. 11, the wrapping model exhibits a deeper and more localized minimum basin, whereas the bridging model samples a broader region of conformational space. While absolute free energies are not directly comparable between these two systems, this qualitative difference suggests that, under our simulation conditions, the wrapping model explores a more convergent conformational ensemble, consistent with visual inspection of the trajectories. We have revised the text accordingly to emphasize this point, while avoiding over-interpretation in terms of absolute thermodynamic stability.

Minor Issues and Clarifications:

1) Line 609: The sentence beginning with “we generated two starting models...” was difficult to follow. While the text refers to two models—wrapping and bridging—the cited figure (Fig. S6) appears to show two different starting models for the wrapping mode only, which differs from what is shown in Fig. 5A. This inconsistency made it unclear whether both the wrapping and bridging models had two starting conformations each, or only one of them did. Clarifying this part of the text, possibly by specifying how many initial models were created and which ones were used in the final analysis, would greatly aid reader understanding.

We thank the reviewer for this helpful comment. To improve clarity, we have revised the description of our modeling workflow in the *Results* section. Specifically, we now clearly state the number of initial models generated. In addition, we have expanded the Supplementary Information to include new figures illustrating the generation of two starting models for each mode—the bridging and wrapping conformations (Supplementary Figs. 8 and 9).

2). Trajectory Selection: Could the authors elaborate on the rationale for choosing a single “representative” trajectory for each model in the final analysis? Given that two independent simulations were performed, it would be more robust to report averaged quantities across both trajectories, or at least provide justification for the representative selection.

We thank the reviewer for pointing out this ambiguity and for raising the question of trajectory selection. For each binding mode (wrapping and bridging), we generated two alternative starting conformations based on the HLP-DNA crystal structures, resulting in four initial models in total. Each was subjected to a 1- μ s MD simulation. The paired replicates for each binding mode exhibited highly consistent behavior. For clarity and to avoid redundancy in the main text, we therefore present one representative trajectory per mode in Fig. 5, while the corresponding replicate analyses are now provided in the Supplemental Information (Supplementary Fig. 10). This clarification has been incorporated into the revised manuscript text.

3). Figure 5 Caption, Line 953: “represent” -> “representing”

This was corrected.

4). Line 292–294 – Protein–DNA Contact Definition: The definition of a protein–DNA contact is not clearly worded. A more precise phrasing would be: “A protein–DNA contact was defined and counted if any protein heavy atom was within 0.4 nm of a DNA heavy atom.” Rewriting this sentence for clarity would make the methodology more accessible, especially to readers less familiar with contact analysis in MD simulations.

This was corrected.

Reviewer #4 (Remarks to the Author):

I co-reviewed this manuscript with one of the reviewers who provided the listed reports. This is part of the Nature Communications initiative to facilitate training in peer review

and to provide appropriate recognition for Early Career Researchers who co-review manuscripts.

Additional revisions implemented independently of reviewer request

METHODS:

Light microscopy imaging and data processing

Previously missing information has now been incorporated into the text, with all changes highlighted in red.

The detailed descriptions of the **Crystallization, data collection and structure determination, Molecular dynamics (MD) simulations, and DNA-binding assays** have been moved to the Supplementary Methods section of the Supplementary Information.

Fig. 6:

We have made minor corrections to the DNA fragment sizes reported in the table corresponding to panel a. Specifically, the value for HLP band 2 was updated from **58** \pm 1 to **59** \pm 1, and the value for HMfB band 1 was corrected from **89** \pm 1 to **88** \pm 1.

In addition, to enhance clarity and ensure direct comparability of the data presented in Fig. 6c, we repeated the experiment using identical protein-to-DNA mass ratios for both HLP and HMfB. Previously, the HLP samples were analyzed with ratios of **0.2, 0.4, 0.5, 0.7, 1.0, 1.5, and 2.0**, whereas HMfB samples were tested with ratios of **0.1, 0.2, 0.4, 0.5, 0.7, 1.0, and 1.5**.

Fig. 8

has been added as a new figure.

Re-numbering of Supplementary Figures:

Supplementary Figures 1a and b have been divided into two separate figures, and four additional Supplementary Figures have been introduced. The updated numbering of all Supplementary Figures is summarized in the table below.

Supplementary Fig.	remarks
1	old S1b
2	old S1a
3	old S2
4	old S3
5	old S4
6	new
7	old S5
8	new
9	old S6
10	new
11	new
12	old S7
13	old S8
14	old S9
15	old S10
16	old S11
17	old S12
18	old S13
19	old S14
20	old S14

Some of the **Figure legends** were updated according to the policies of *Nature Communications*. Changes are marked in red.

References used in response to reviewers

- Faltejsková, K., Jakubec, D. & Vondrášek, J. Hydrophobic amino acids as universal elements of protein-induced DNA structure deformation. *Int. J. Mol. Sci.* 21, 3986 (2020). <https://doi.org/10.3390/ijms21113986>
- Freeman, G. S., Lequieu, J., Hinckley, D. M., Whitmer, J. K. & de Pablo, J. J. DNA shape dominates sequence affinity in nucleosome formation. *Phys. Rev. Lett.* 113, 168101 (2014). <https://doi.org/10.1103/PhysRevLett.113.168101>
- Rohs, R., West, S. M., Sosinsky, A., Liu, P., Mann, R. S. & Honig, B. The role of DNA shape in protein–DNA recognition. *Nature* 461, 1248–1253 (2009). <https://doi.org/10.1038/nature08473>
- Virstedt, J., Berge, T., Henderson, R. M., Waring, M. J. & Travers, A. A. The influence of DNA stiffness upon nucleosome formation. *J. Struct. Biol.* 148, 66–85 (2004). <https://doi.org/10.1016/j.jsb.2004.03.007>
- West, S. M., Rohs, R., Mann, R. S. & Honig, B. Electrostatic interactions between arginines and the minor groove in the nucleosome. *J. Biomol. Struct. Dyn.* 27, 861–866 (2010). <https://doi.org/10.1080/07391102.2010.10508587>
- Winogradoff, D., Zhao, H., Dalal, Y. & Papoian, G. A. Shearing of the CENP-A dimerization interface mediates plasticity in the octameric centromeric nucleosome. *Sci Rep* 5, 17038, doi:10.1038/srep17038 (2015).
- Yakovchuk, P., Protozanova, E. & Frank-Kamenetskii, M. D. Base-stacking and base-pairing contributions into thermal stability of the DNA double helix. *Nucleic Acids Res.* 34, 564–574 (2006). <https://doi.org/10.1093/nar/gkj454>
- Zhao, H., Winogradoff, D., Dalal, Y. & Papoian, G. A. The Oligomerization Landscape of Histones. *Biophys J* 116, 1845–1855, doi:10.1016/j.bpj.2019.03.021 (2019).
- Zuccheri, G., Scipioni, A., Cavaliere, V., Gargiulo, G., De Santis, P. & Samorì, B. Mapping the intrinsic curvature and flexibility along the DNA chain. *Proc. Natl Acad. Sci. USA* 98, 3074–3079 (2001). <https://doi.org/10.1073/pnas.051631498>

Response to Reviewer 3

Our response is in blue letters.

Most of my concerns have been addressed. Here are some minor issues regarding the 5th response:

1. SI figures like Fig S11 lacks of unit for the color bar.

We have added the units and descriptive titles to the color bars in Supplementary Figures 11 (now Suppl. Fig. 12), 13 (now Suppl. Fig. 14), and 14 (now Suppl. Fig. 15).

2. According to the SI Methods section, the trajectories were aligned to a common reference structure. However, it appears that the two landscapes in Figure S11 were computed using different reference structures, as the two binding modes are initially distinct and the authors did not specify that the same reference was used. If this is the case, the principal components (PC1 and PC2) derived from the two binding modes are not directly comparable, as they represent different coordinate bases. Therefore, comparing these two free energy landscapes side by side may not be meaningful.

To provide a more interpretable comparison, I would suggest projecting the trajectories onto biophysically meaningful parameters—such as the number of contacts between the DNA and the protein, the RMSD, or the radius of gyration (Rg) of the DNA—some used in the main text. It would allow for a more consistent and physically interpretable comparison between the two binding modes. We agree with this concern and thank the reviewer for the helpful observation. As recommended, we recomputed the two free-energy landscapes using biophysically meaningful parameters—the RMSD of the DNA relative to the reference structure and DNA radius of gyration—to provide a consistent and physically interpretable comparison between the two binding modes. The revised panels are now presented as Supplementary Figure 12.